# Enhancer decommissioning by MLL4 ablation elicits dsRNA-interferon signaling and GSDMD-mediated pyroptosis to potentiate anti-tumor immunity

Hanhan Ning[1,6], Shan Huang[1,6], Yang Lei [2,6], Renyong Zhi[1], Han Yan[1], Jiaxing Jin[1], Zhenyu Hu[2], Kaimin Guo[2], Jinhua Liu[1], Jie Yang [3], Zhe Liu [4], Yi Ba [5], Xin Gao [2] ✉ & Deqing Hu [1,2,5] ✉

Enhancer deregulation is a well-established pro-tumorigenic mechanism but whether it plays a regulatory role in tumor immunity is largely unknown. Here, we demonstrate that tumor cell ablation of mixed-lineage leukemia 3 and 4 (MLL3 and MLL4, also known as KMT2C and KMT2D, respectively), two enhancer-associated histone H3 lysine 4 (H3K4) mono-methyltransferases, increases tumor immunogenicity and promotes anti-tumor T cell response. Mechanistically, MLL4 ablation attenuates the expression of RNA-induced silencing complex (RISC) and DNA methyltransferases through decommissioning enhancers/super-enhancers, which consequently lead to transcriptional reactivation of the double-stranded RNA (dsRNA)-interferon response and gasdermin D (GSDMD)-mediated pyroptosis, respectively. More importantly, we reveal that both the dsRNA-interferon signaling and GSDMD-mediated pyroptosis are of critical importance to the increased anti-tumor immunity and improved immunotherapeutic efficacy in MLL4-ablated tumors. Thus, our findings establish tumor cell enhancers as an additional layer of immune evasion mechanisms and suggest the potential of targeting enhancers or their upstream and/or downstream molecular pathways to overcome immunotherapeutic resistance in cancer patients.

Cancer immunotherapies through immune checkpoint blockade (ICB) have manifested tremendous clinical efficacy in a fraction of patients with diverse types of cancers[1]. However, a substantial percentage of patients either fail to respond or develop therapeutic resistance with an ultimate consequence of cancer relapse[1,2]. Thus, a thorough understanding of the molecular and cellular mechanisms behind tumor immune response and evasion is expected to increase the immunotherapeutic efficacy and achieve a better prognosis in cancer patients with primary or acquired resistance to ICB therapies. Multiple tumor cell-intrinsic mechanisms, including tumor mutational load[3–6], defects

in interferon response, and major histocompatibility complex I (MHC I) antigen presentation[7–10], dysregulated metabolic networks,[11–13] and aberrant oncogenic signaling pathways[14,15], etc., have recently been identified to regulate the tumor immune landscapes and immunotherapeutic resistance. Despite these mechanistic findings, it remains incompletely understood whether there are additional layers of regulation that underlie the tumor immune regulation, either function individually or in cooperation with the currently identified mechanisms.

Perturbation of epigenetic mechanisms through genetic alternations or aberrant expression of chromatin regulators plays an

instrumental role in controlling tumor-cell properties and has become attractive therapeutic targets for cancer treatment. More recently, chromatin regulators gained increasing attention in the field of cancer immunology as they were also noted to regulate tumor immune landscape and thus affect the clinical response to checkpoint blockade therapies. For example, CRISPR/Cas9 screening identified a crucial role for polycomb repressive complex 2 (PRC2), an H3K27 methylase, in silencing the expression of MHC class I (MHC-I) to enable tumor immune evasion[16,17]. Cooperation between PRC2 and DNMT1-mediated DNA methylation was shown to suppress the expression of T helper 1 (Th1)-type chemokines CXCL9 and CXCL10 to impede tumor infiltration of CD8[+] T cells[18]. Targeting DNA and histone modifiers leads to depression of retroviral and transposable elements to promote interferon signaling and tumor immunogenicity through induction of double-stranded RNA (dsRNA) stress or viral-encoded neoantigens[19–23]. These studies collectively demonstrate the crucial roles of chromatin modifiers in the transcriptional regulation of tumor immunity and clinical response to ICB therapies.

Enhancers dictate and sustain the expression of lineage-determining genes during embryonic development[24]. As key regulators of gene transcription, enhancer activity is tightly controlled by transcription factors and chromatin regulators. Monomethyl histone H3 lysine 4 (H3K4me1), primarily catalyzed by MLL3 and MLL4 branches of the COMPASS (Complex of Proteins Associated with Set1) family of H3K4 methylases, is an epigenetic signature of eukaryotic enhancers and commonly utilized to pinpoint the genomic position of enhancers or to annotate enhancer states when combined with additional histone marks[25–27]. Deregulation of enhancer activity is known to be a primary cause for the development of various diseases, including cancer[24]. The past decade of cancer genome sequencing efforts has led to the realization that enhancer DNA itself and the enhancer-associated chromatin regulators, including, but not limited to MLL3 and MLL4, are frequently mutated in various types of human cancers[28–30]. Cancer-associated genetic lesions in MLL3 and MLL4 are predominantly nonsense and frameshift mutations that lead to protein truncation and loss of function[31]. Based on the biochemical function we and others revealed, we previously proposed that enhancer malfunction may serve as the underlying mechanisms and driving force for the aggressive behaviors of tumor cells bearing MLL3 or MLL4 mutations, which have been corroborated in human cancer cell lines and murine genetic models of multiple cancers[32–36]. However, whether the tumor cell-intrinsic transcriptional enhancers, in particular, MLL3 and MLL4-regulated enhancers, are involved in tumor immune response and therapeutic resistance to checkpoint blockades remain incompletely understood.

Pyroptosis is a programmed necrotic cell death mediated by proteolytic cleavage of the Gasdermin (GSDM) family of pore-forming proteins that comprise GSDMA1-3, GSDMC 1-4, GSDMD, GSDME, and DFNB59 (official name known as PJVK) in mouse[37]. The GSDMs are normally self-inhibited through the intramolecular association between the N-terminal pore-forming domain and the C-terminal fragment[38,39]. Upon bacterial infection and/or encountering danger signals, cleavage in the linker region unleashes the pore-forming activity of the N-terminal domain on the plasma membrane and leads to the release of proinflammatory molecules, such as cytokines IL-1β and IL-18 and cellular alarmins ATP and HMGB1, into extracellular space[40]. Current studies suggest a tumor-suppressive function of pyroptosis, as epigenetic silencing and genetic mutations have been found in some of GSDMs in cancer patients[40]. In line with this notion, the latest studies revealed that tumor-cell pyroptosis induced by killer lymphocyte-derived granzymes cleavage of GSDMB and GSDME is immunostimulatory and suppresses tumor progression by shaping the immune landscape and promoting lymphocyte activation in the tumor microenvironment[41,42]. However, whether tumor-cell pyroptosis is generally involved in anti-tumor immune response, or more

specifically whether the pyroptotic processes elicited by other members of the GSDM family control anti-tumor immunity and how they are epigenetically and transcriptionally regulated remains largely unknown.

Here, we present evidence that tumor-cell enhancer H3K4 mono-methylases MLL3 and MLL4 suppress T cell-mediated cytotoxicity and promote tumor immunosuppression in syngeneic murine models of multiple cancer types. Mll3 or Mll4 deletion leads to robust transcriptional reactivation of interferon response and pyroptotic pathway in tumor cells. Mechanistic studies demonstrated that MLL4 loss weakens the strength of both canonical and super enhancers and thereby attenuates the expression of their adjacent genes, including the RNA-induced silencing complex (RISC) component AGO2 and DNA methyltransferases DNMT1 and DNMT3A. The negative effects on the expression of AGO2 and DNA methyltransferases by MLL4 loss promote double-stranded RNA (dsRNA) stress and derepress Gsdmd and inflammatory caspases to trigger interferon response and pyroptosis, which account for the increased anti-tumor immunity and immunotherapeutic efficacy induced by tumor cell-intrinsic MLL4 ablation. Our findings support a general function of tumor-cell pyroptosis in promoting anti-tumor immunity and reveal the functional link between regulation of MLL4-dependent enhancers and transcriptional induction of interferon signaling and GSDMD-mediated pyroptosis in tumor cells for immunomodulation.

## Results

### Identification and validation of tumor cell-intrinsic MLL3 and MLL4 as suppressors for CD8[+] T-cell activation and cytotoxicity in vitro

To identify regulators that determine anti-tumor T-cell response, we first reanalyzed a recently published genome-scale CRISPR/Cas9 screen dataset that sought to identify tumor cell-intrinsic modulators of antigen-specific OT-1 and Pmel-1 CD8[+] T-cell cytotoxicity in vitro[43]. Mll3 and Mll4 stand out as top hits, to suppress the cytotoxicity of antigen-specific CD8[+] T cells in vitro (Fig. 1a). Analysis of an additional CRISPR/Cas9 genetic screen for epigenetic regulators of anti-tumor immune response in vivo revealed significant depletion of Mll4 sgRNAs in both B16 melanoma and Lewis lung carcinoma (LLC1) tumors, as well as markedly reduced representation of both Mll3 and Mll4 sgRNAs in LLC1 tumors in immunocompetent mice as compared to the corresponding tumors in immunocompromised NSG mice (Supplementary Fig. 1a), indicating an immunosuppressive function of tumor cell-intrinsic MLL3 and MLL4 in vivo as well[23]. Initial findings from both genetic screens inspired us to evaluate the involvement of all members of the COMPASS family of H3K4 methyltransferases in tumor immunity. To this end, we depleted H3K4 methyltransferase individually in B16 cells that express exogenous Ova antigen (Supplementary Fig. 1b, c) and titrated the effector to target (E/T) ratios for optimal OT-1 T cytotoxicity based on the lactate dehydrogenase (LDH) release assay (Supplementary Fig. 1d). The targeted screen revealed that knockdown of Mll3 and Mll4, but not other H3K4 methyltransferases, markedly increases the cytotoxic potential of antigen-specific T cells as shown by increased release of LDH from Mll3- and Mll4-depleted cells as compared to the control cells (Fig. 1b). To further validate their immune suppressive function, we knocked out Mll3 and Mll4 individually in B16 cells by CRISPR/Cas9 (Supplementary Fig. 1e–k) and observed that both MLL3 and MLL4 depletion significantly increase LDH release and B16 cell death when co-cultured with OT-1 CD8[+] T cells (Fig. 1c, d).

Phenotypic and functional characterization revealed that the viability, proliferation, activation, and cytotoxic potential of OT-1 CD8[+] T-effector cells are all increased when they are co-cultured with Mll3- and Mll4-deficient B16-Ova cells as compared to the co-culture with control target cells (Fig. 1e–h). Knockdown of Mll3 and Mll4 in target B16-Ova cells also leads to increased proliferative potential, more

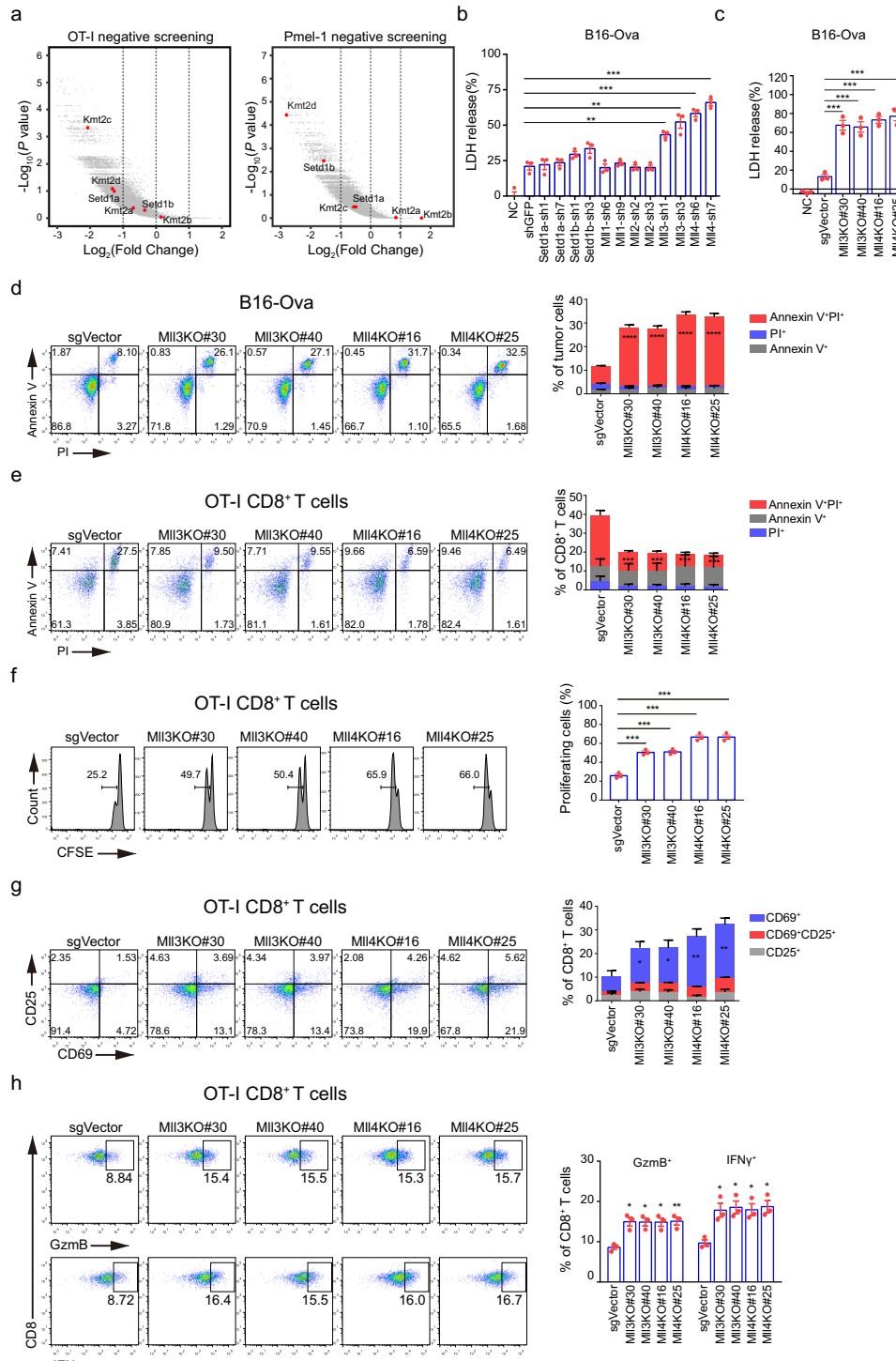

**Fig. 1 | Loss of *Mll3* and *Mll4* in tumor cells stimulates activation and cytotoxicity of antigen-specific CD8[+] T cells in vitro. a** Volcano plots showing in vitro CRISPR/Cas9 screening of tumor-cell-intrinsic factors that regulates cytotoxicity of OT-I and Pmel-1 CD8[+] T cells. Genes were plotted based on mean log2 fold change of gRNA counts compared to control selection. H3K4 methyltransferases of the COMPASS family were highlighted in red. Datasets for OT-I and Pmel-1 screening were from Pan et al.[43] and reanalyzed in this study. **b** B16-Ova tumor cells depleted for the indicated H3K4 methyltransferase were incubated with OT-I CD8[+] T cells at an effector to target (E/T) ratio of 10:1 for 24 h. OT-I T-cell-mediated killing was determined by lactate dehydrogenase (LDH) release-based cell death analyses.

**c, d** Two independent *Mll3* or *Mll4* knockout (KO) B16-Ova clonal cells were co-cultured with OT-I CD8[+] T cells for 24 h and tumor-cell death was analyzed by LDH release (**c**) and Annexin V/PI staining (**d**). **e, f** The experiment was conducted as in (**b**) followed by cell death (**e**) and proliferation (**f**) analyses of OT-I CD8[+] T cells by flow cytometry. **g, h** The experiment was performed as in (**b**) followed by activation (**g**) and cytotoxicity (**h**) analyses of OT-I CD8[+] T cells by flow cytometry. Quantification in (**b–h**) was shown as mean ± SEM from three biological replicates. Statistical significance was determined by MaGeCK (Model-based Analysis of Genome-wide CRISPR-Cas9 Knockout) (**a**) and two-tailed unpaired t test (**b–h**). *$P < 0.05$; **$P < 0.01$; ***$P < 0.001$.

viability, and elevated levels of IFNγ and GzmB in antigen-specific CD8-T cells (Supplementary Fig. 1l–n). Moreover, we found tumor-cell loss of MLL3 or MLL4 promotes survival, proliferation, and cytotoxicity of antigen-nonspecific CD8[+] T cells pre-activated by anti-CD3/CD28 antibodies as well (Supplementary Fig. 1o–q). Collectively, these findings strongly suggest that tumor cells may exploit MLL3 and MLL4 to suppress activation and cytotoxic activity of CD8[+] T cells to evade immune cell killing.

### MLL3 or MLL4 ablation promotes antigen-specific T-cell activation and suppresses tumor progression in immunocompetent mice

Our in vitro findings point to a potentially critical function of tumor-cell-intrinsic MLL3 and MLL4 in regulating anti-tumor immunity. To test this notion, we inoculated subcutaneously the wild-type and mutant B16 melanoma cells deleted for *Mll3*, *Mll4*, or both (DKO) into immune-competent syngeneic C57BL/6J mice or immune-compromised BALB/c nude mice and *Rag1*[−/−] mice. Ablation of MLL3, MLL4, or both dramatically suppresses tumor growth and reduces tumor burden in immunocompetent mice and thus conferring a marked survival benefit to these mice (Fig. 2a–c). No significant difference in tumor growth rate and tumor burden was observed between the wild-type or mutant tumor cells that were engrafted into immune-deficient mice (Supplementary Fig. 2a–d), indicating the dependence of the intact murine immune response for tumor-suppressive effects of MLL3 or MLL4 ablation. To further corroborate our notion, we knocked down *Mll3* or *Mll4* in murine lung cancer LLC and colon cancer MC38 cell lines and inoculated them into immune-competent syngeneic C57BL/6J mice. Depletion of MLL3 or MLL4 significantly attenuates the tumor growth rate and confers survival advantages to these mice as compared to control tumors (Supplementary Fig. 2e–h). Lung is the most common invading site of metastatic melanoma cells. We found loss of MLL3 and MLL4 completely abolishes the metastatic potential of melanoma cells when intravenously injected into immunocompetent mice (Supplementary Fig. 2i), indicating potential involvement of immune response for inhibition of melanoma metastasis and colonization as well.

Given the predominant function of cytotoxic CD8[+] T cells in tumor immune rejection[43], we examined the role of antigen-specific CD8[+] T cells in immune suppression of *Mll3*[−/−] and *Mll4*[−/−] melanoma cells. *Rag1*[−/−] mice were implanted with non-targeting control, *Mll3*[−/−] or *Mll4*[−/−] B16-Ova cells, and then adoptively transferred with Ova-specific CFSE-labeled OT-I CD8[+] T cells (Fig. 2d). In contrast to the comparable tumor growth rate in immuno-deficient mice (Supplementary Fig. 2a–d), OT-I CD8[+] T-cell transfer results in much slower progression and reduced tumor burden of *Mll3*[−/−] or *Mll4*[−/−] B16 cells in *Rag1*[−/−] mice as compared to the control tumors, indicating ablation of MLL3 or MLL4 promotes the anti-tumor function of adoptive CD8[+] T cells (Fig. 2e, f). Bioluminescent imaging of tumors and mouse organs revealed much stronger fluorescent signals in *Mll3*[−/−] or *Mll4*[−/−] melanomas than in control tumors, while equivalent fluorescent intensity was detected in livers of mice engrafted with control, *Mll3*[−/−] or *Mll4*[−/−] B16 cells (Fig. 2g). To more precisely evaluate the effects of tumor-cell MLL3 or MLL4 ablation on tumor infiltration and activity of CD8[+] T cells, we inoculated C57BL/6J mice with control or mutant B16 cells and observed a significantly higher frequency of CD3[+] and CD8[+] T cells in *Mll3*[−/−] or *Mll4*[−/−] melanomas than in control tumors (Fig. 2h, i). In addition, we found that CD8[+] T cells in *Mll3*[−/−] or *Mll4*[−/−] melanomas are more active, divide faster, exhibit less cell death, and express a higher level of effector molecules than in control tumors (Fig. 2j–m). These results strongly indicate that tumor-cell loss of MLL3 or MLL4 promotes anti-tumor T-cell function in vivo.

To determine the clinical relevance of our findings, we analyzed TCGA (The Cancer Genome Atlas) RNA-seq datasets on human cancer patients and revealed inverse correlations for the expression of *MLL3*

or *MLL4* with *CD3* and *CD8* mRNA levels in diverse cancer types (Supplementary Fig. 2j, k). Expression of *MLL3* or *MLL4* is also negatively associated with *GZMA*, *GZMB*, and *IFNG* mRNA levels in a variety of human cancer types, highlighting increased immune cytotoxicity towards MLL3- and MLL4-low tumors (Fig. 2n and Supplementary Fig. 2l, m). Elevated immune cytotoxicity in MLL3- and MLL4- low tumors not only results from the higher degree of T-cell infiltration but is also contributed by the increased cytotoxicity of infiltrated CD8[+] T cells as the inverse correlation between the expression of MLL3 or MLL4 with the ratio of GZMB/CD8 transcripts was observed in most TCGA tumor types (Fig. 2o), which is in line with our findings in murine melanoma. Moreover, a trend of substantial survival benefit was observed in patients with MLL3- or MLL4-low melanoma as compared to patients bearing MLL3- or MLL4-high melanoma when a high degree of CD8[+] T-cell infiltration occurs (Fig. 2p). Pan-cancer analyses of TCGA tumors reveal a higher abundance of total and CD8[+] T cells as well as increased immune cytotoxicity in *MLL3*- or *MLL4*-mutated tumors compared to tumors devoid of changes in the respective gene (Fig. 2q, r, and Supplementary Fig. 2n, o). Furthermore, we found tumors with *MLL4* mutation are associated with a better response to PD1-PD-L1 blockade immunotherapies in metastatic and urothelial cancer patients[44–46] (Fig. 2s). Collectively, these data suggest that loss of tumor-cell MLL3 or MLL4 may promote the activation of CD8[+] T cells and elicits potent anti-tumor immunity in human as well.

### MLL3 or MLL4 loss promotes cytosolic dsRNAs stress to elicit transcriptional induction of interferon response

To determine how MLL3 or MLL4 suppresses anti-tumor T-cell immunity, we performed total RNA-seq analyses of tumor cells sorted out from C57BL/6J mice inoculated with GFP-labeled non-targeting control, *Mll3*[−/−] or *Mll4*[−/−] B16 cells (Fig. 3a, b, and Supplementary Fig. 3a, b). Gene set enrichment analysis (GSEA) of differentially expressed genes revealed robust activation of both innate antiviral immune response and adaptive immunity, including response to type I and type II interferon, T-cell-mediated immunity, etc. in *Mll3*[−/−] or *Mll4*[−/−] tumor cells (Fig. 3c and Supplementary Fig. 3c). Expression of many genes that mediate interferon production, transduce extracellular interferon signaling or are induced directly as interferon-stimulated genes (ISGs) are markedly upregulated in *Mll4*[−/−] B16 cells and diverse human cancer cell lines knocked down for *MLL4* expression (Fig. 3d and Supplementary Fig. 3d). Some of these interferon responsive genes are also transcriptionally induced in *Mll3*[−/−] tumor cells (Supplementary Fig. 3e). Type I and type II interferons are well-known inducers of MHC I-dependent adaptive immune response[47,48]. In line with this, MLL4 depletion dramatically increases the expression of genes encoding MHC I machineries and components in antigen processing and presentation pathways in both murine B16 melanoma cells and a few examined human cancer cell lines (Fig. 3e and Supplementary Fig. 3f), most of which are also transcriptionally induced in *Mll3*[−/−] B16 tumor cells as well (Supplementary Fig. 3g). Consistent with the gene expression data, flow cytometry analysis revealed that ablation of *Mll3* or *Mll4* markedly increases the tumor cell-surface levels of both free and Ova-bound MHC I (Fig. 3f, g). Furthermore, expression of multiple T-chemoattractants, including CXCL9 and CXCL10[49], is also markedly induced in *Mll4*[−/−] tumor cells and bulk tumors (Supplementary Fig. 3i, j). Together, these results suggest that activation of multiple immune signaling pathways may underlie the stimulating effects of tumor cell-intrinsic MLL3 and MLL4 ablation on adaptive anti-tumor T-cell immune response.

Cytosolic nucleic acids, including both dsRNAs and dsDNAs, are potent stimulators of intracellular interferon response and tumor immunogenicity[21,22]. We found expression of cytosolic dsRNAs and dsDNAs sensors is markedly increased in *Mll3*[−/−] or *Mll4*[−/−] tumor cells (Fig. 3h and Supplementary Fig. 3h) and depleting two of the dsRNAs sensors, MDA5 and RIG-I (encoded by *Ddx58*), partially rescues the

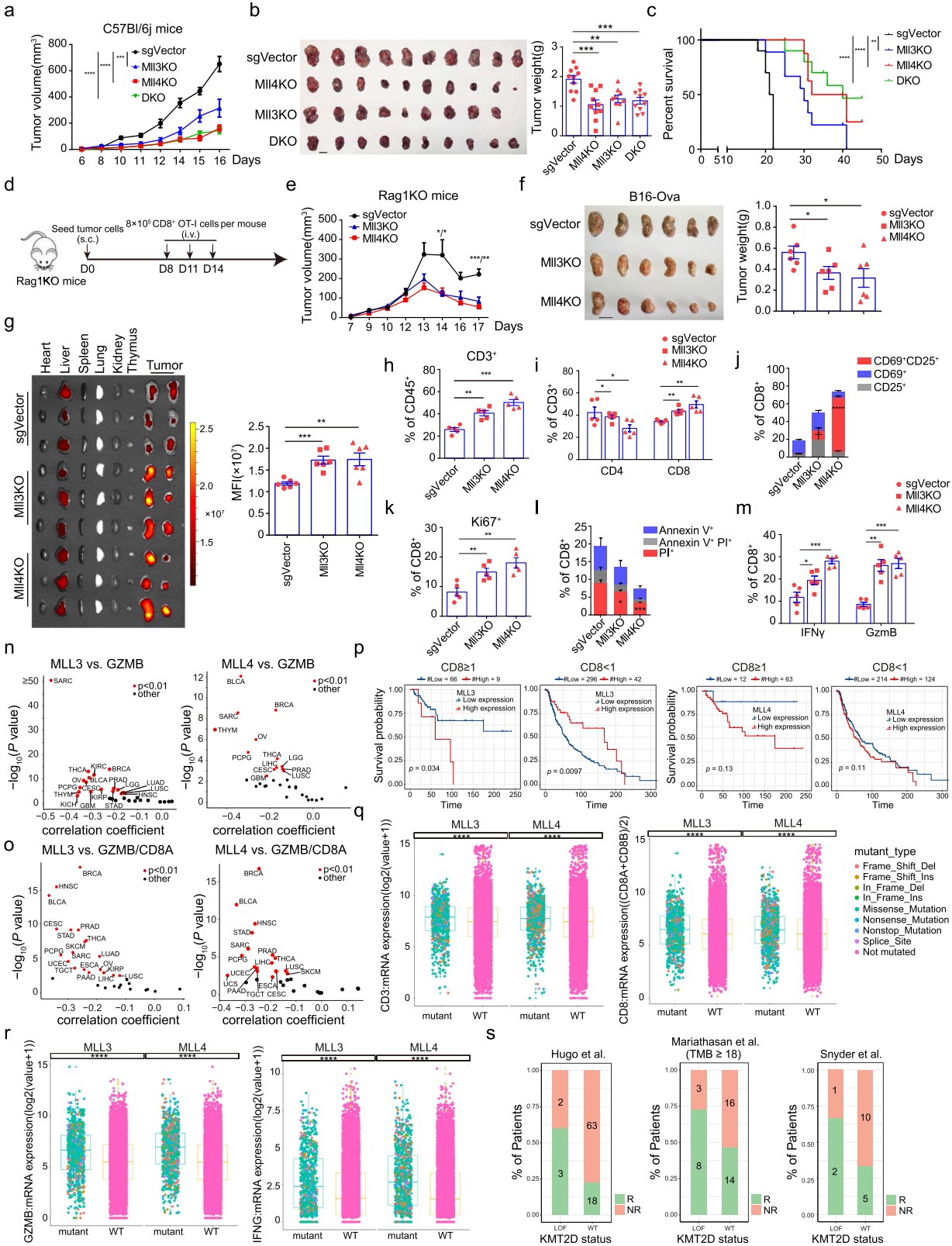

tumor growth defects and abrogates the survival advantages of C57BL/6J mice implanted with *Mll4*[−/−] melanoma cells (Fig. 3i, j), indicating the necessity of innate dsRNAs sensing pathways in the efficient immune clearance of *Mll4*[−/−] tumors in immune-competent mice. Bidirectional transcripts from endogenous retroviral elements (ERVs) and other retrotransposons are a major source of immunogenic dsRNAs for stimulating interferon response[50–52]. Transcriptomic analyses reveal that

deletion of *Mll3* or *Mll4* causes a remarkable increase in the level of bidirectional ERVs transcripts (Fig. 3k and Supplementary Fig. 3k), which may contribute to the accumulation of cytosolic dsRNAs in *Mll4*[−/−] B16 tumor cells and human cancer cells depleted for MLL4 expression as demonstrated by immunofluorescent staining using a dsRNAs-specific J2 antibody (Fig. 3l and Supplementary Fig. 3l, m). Cytosolic level of dsRNAs is not only determined by the degree of ERVs

**Fig. 2 | Tumor-cell loss of *Mll3* or *Mll4* enhances CD8+ T-cell function and promotes tumor immunosuppression in vivo. a, b** Tumor growth curve (**a**) and tumor images and weight quantification at the experimental endpoint (**b**) (mean ± SEM, *n* = 10). **c** Kaplan–Meier survival curves for mice in (**a**). **d** Diagram for the adoptive transfer of CFSE-labeled OT-I CD8+ T cells into *Rag1*−/− mice (*n* = 3) bearing indicated B16-Ova tumors. **e, f** Tumor growth curve (**e**) and tumor images and weight quantification at the time of mice sacrifice (**f**) (mean ± SEM, *n* = 6). **g** Tissue distribution and quantification of fluorescent signal 3 days post the last OT-I T transfer as outlined in (**d**). Fluorescent intensity in tumors was shown as mean ± SEM (*n* = 6). **h, i** Frequency for infiltrated total (**h**), CD4+ and CD8+ T cells (**i**) in indicated B16 tumors of C57BL/6J mice 14 days post implantation (mean ± SEM, *n* = 5). **j–m** Mice were treated as in (**h**). Activation (**j**), proliferation (**k**), apoptotic states (**l**), and cytotoxicity (**m**) of CD8+ T cells were shown as mean ± SEM (*n* = 5). **n, o** Volcano plots showing the Spearman's correlation and estimated significance for indicated pairs in RNA-seq datasets across TCGA cancer types[88]. Each dot represents a cancer type and significant correlations are highlighted in red. **p** Kaplan–Meier survival curves for SKCM (Skin Cutaneous Melanoma) patients with differential mRNA levels of *MLL3* or *MLL4* and *CD8* [(CD8A + CD8B)/2]. For CD8 ≥ 1, Mll3low (*n* = 66), Mll3high (*n* = 9), Mll4low (*n* = 12), Mll4high (*n* = 63); For CD8 < 1, Mll3low (*n* = 296), Mll3high (*n* = 42), Mll4low (*n* = 214), Mll4high (*n* = 124). **q, r** Boxplots of total and CD8+ T cells infiltration (**q**) and expression of cytotoxic effectors (**r**) in TCGA pan-cancer patients bearing wild-type or genetically altered *MLL3* or *MLL4*. Center line, median; box bounds, upper and lower quartiles; whiskers, 1.5× IQR (interquartile range). **s** Clinical response of PD-L1-PD1 blockade therapy in melanoma and urothelial cancer patients bearing wild-type or genetically inactivated *MLL4*[44–46]. R responder, NR nonresponder. Statistical significance was determined by two-tailed unpaired t (**b, f, g, h–m**), two-way ANOVA (**a, e**), log-rank (Mantel–Cox) test (**c**), cor.test (**n, o**), log-rank test (**p**) or two-sided Wilcoxon test (**q, r**).*P < 0.05; **P < 0.01; ***P < 0.001.

transcription but also constrained by the action RISC complex, which comprises DICER, TRBP2, RHA, and members of the AGO family (Fig. 3m)[21,53]. Intriguingly, we found MLL4 depletion leads to a prominent reduction in both transcript and protein levels of AGO2 in B16 tumor cells and a few examined human cancer lines (Fig. 3n and Supplementary Fig. 3n, o). To explore a potential broad regulatory function of MLL4 in AGO2 expression in human tumor cells, we analyzed the expression correlation between the mRNA levels of *MLL4* and *AGO2* in the TCGA human tumors and the CCLE) human tumor-cell lines. *AGO2* expression positively correlates with the *MLL4* mRNA level in most TCGA tumor types, including skin cutaneous melanoma (SKCM), as well as in CCLE tumor-cell lines when correlative analyses were conducted using either all cell lines or cell lines grouped according to their tissue of origin (Fig. 3o, p).

A previous study reported that AGO2 depletion elevates the abundance of dsRNAs derived from retrotransposons and results in the induction of interferon and ISGs in human breast cancer cells[21], suggesting potential roles of AGO2 in tumor immunity and anti-tumor immune response. Bioinformatic analyses reveal that AGO2 expression is negatively correlated with the mRNA levels of *CD3*, *CD8A*, and *GZMA* in a variety of TCGA human tumor types (Fig. 3q, r), indicating AGO2-low tumors have a higher abundance of total and CD8+ T cells and stronger immune cytotoxic activity. In addition to more intratumoral CD8+ T cells, the increased immune cytotoxicity in AGO2-low tumors may also be contributed by stronger cytotoxicity of infiltrated CD8+ T cells as inverse correlations were observed for the expression of *AGO2* with the ratio of GZMA/CD8A mRNA level in diverse TCGA tumor types as well (Fig. 3s). Therefore, these findings prompted us to investigate the potential role of AGO2 downregulation in immune suppression of *Mll4*−/− melanomas. Re-introduction of exogenous *Ago2* partially rescues tumor growth defects in immunocompetent mice (Fig. 3t and Supplementary Fig. 3p). Flow cytometry analysis revealed that ectopic expression of *Ago2* decreases the cell-surface level of MHC I and infiltration and cytotoxicity of CD8+ T cells in *Mll4*−/− melanomas (Fig. 3u, v, and Supplementary Fig. 3q), which probably underlie the rescuing effects of exogenous *Ago2* expression on *Mll4*−/− melanoma progression in vivo. An elevated level of *PD-L1* mRNA level was also noted in several human cancer cell lines upon MLL4 depletion (Supplementary Fig. 3r). Altogether, these data indicate that accumulation of retroviral elements-derived dsRNAs and consequent induction of interferon signaling are at least partially responsible for increased tumor immunity of *Mll4*−/− melanoma cells.

### *Mll3* or *Mll4* ablation induces transcriptional priming of the GSDMD-mediated pyroptotic pathway

Besides the enrichment of antiviral interferon response and T-cell-mediated adaptive immunity (Fig. 3c and Supplementary Fig. 3c), GSEA analyses also revealed pronounced induction of genes involved in pyroptosis, including pyroptotic executioner *Gsdmd* and the inflammatory *Casp1* and *Casp11* in *Mll3*−/− and *Mll4*−/− melanoma cells (Fig. 4a and Supplementary Fig. 4a). Transcriptional induction of pyroptotic genes is also noticed in *Mll3*−/− and *Mll4*−/− bulk tumors (Fig. 4b and Supplementary Fig. 4b). Inflammatory caspases-mediated cleavage of GSDMD after residue Asp276 releases its auto-inhibition and produces N-terminal and C-terminal fragments of ~31 kDa and 22 kDa, respectively[38]. We found levels of both full-length and N-terminal fragment of GSDMD are elevated in *Mll3*−/− and *Mll4*−/− bulk tumors (Fig. 4c and Supplementary Fig. 4c, d), indicating that *Mll3* and *Mll4* deletion induces GSDMD expression and could transcriptionally prime tumor cells for pyroptosis. To assess the relevance of these findings in human cancers, we analyzed the expression correlation of *MLL3* and *MLL4* with *GSDMD* across TCGA human tumors. *GSDMD* expression is negatively correlated with the mRNA levels of *MLL3* and *MLL4* in nearly all TCGA human tumor types (Fig. 4d and Supplementary Fig. 4e), suggesting a general function of MLL3 and MLL4 in the control of GSDMD expression in human tumors. Knockdown of *MLL4* causes a notable increase in GSDMD level in three out of four selected human cancer cell lines with distinct tissue origins (Fig. 4e).

Latest studies reported that GSDMB and GSDME in tumor cells are proteolyzed by granzymes of killer lymphocytes to trigger pyroptosis[41,42]. These findings led us to explore if the GSDMD cleavage we observed in bulk tumors is triggered by cytotoxic lymphocytes as well. To this end, we ectopically expressed N-terminally HA-tagged wild-type and D276A mutant GSDMD in B16-Ova cells (Supplementary Fig. 4f). Co-culture of these target cells with OT-I CD8+ T cells leads to the appearance of a ~ 34 kDa cleavage fragment and increased secretion of HMGB1, an immune stimulant of dendritic cell (DC) activation, in wild-type but not the D276A mutant GSDMD-reconstituted B16-Ova cells (Fig. 4f)[54]. Expression of exogenous wild-type but not D276A mutant GSDMD leads to the appearance of giant membrane balloon, extracellular release of LDH and preloaded calcein AM into supernatants, and increased cell death in B16-Ova cells (Fig. 4g, h, and Supplementary Fig. 4g), indicating immune attack by antigen-specific cytotoxic T lymphocytes induces GSDMD cleavage and triggers pyroptotic cell death in tumor cells expressing exogenous GSDMD.

GSDME-mediated pyroptosis attenuates tumor growth and enhances anti-tumor immune response by shaping the immune landscape and state in the tumor microenvironment (TME)[41]. To determine whether GSDMD-induced tumor-cell pyroptosis augments tumor immunity as well, we first evaluated the correlation of *GSDMD* mRNA level with the abundance and cytotoxicity of T lymphocytes in TME across TCGA tumor types. *GSDMD* transcript level positively correlates with the mRNA levels of *CD3*, *CD8A*, *GZMA*, *GZMB*, *PRF1*, and the mRNA ratio of *GZMA* or *GZMB* to *CD8A* in the vast majority of TCGA tumor types (Fig. 4i–k and Supplementary Fig. 4h), indicating higher expression of *GSDMD* is associated with more infiltration and stronger cytotoxic activity of CD8+ T cells. Ectopic expression of wild-type but not D276A mutant GSDMD into B16 cells greatly suppresses tumor

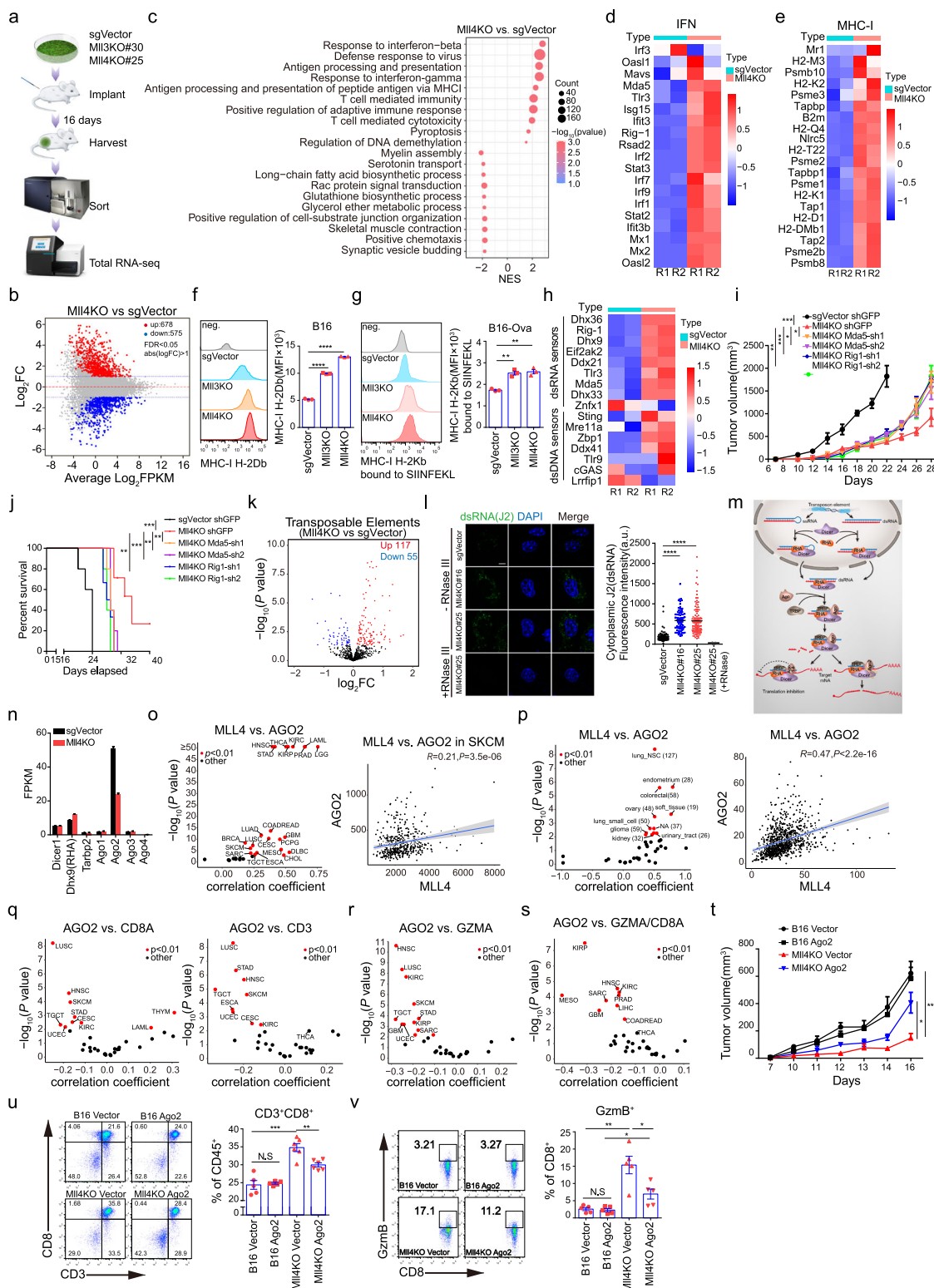

growth and thus reduces the tumor burden in C57BL/6J mice (Fig. 4l, m). The suppressive effects of exogenous GSDMD expression on tumor progression were not noticed when B16 cells were implanted in immune-deficient *Rag1*⁻/⁻ mice (Supplementary Fig. 4i, j), highlighting the requirement of an intact immune system for tumor suppression by GSDMD. In line with these data, we found ectopic expression of wild-type GSDMD increases the intratumoral frequency of both total and CD8⁺ T cells and promotes activation and cytotoxic capacity of infiltrated CD8⁺ T cells (Fig. 4n–p and Supplementary Fig. 4k). In addition, analysis of the TCGA human melanoma patient cohort revealed that

high expression of *GSDMD* is associated with a significant overall survival benefit (Supplementary Fig. 4l). Taken together, these data indicate that GSDMD-mediated pyroptosis enhances tumor immunity and promotes tumor immunosuppression in both murine and a variety of human cancers.

## Transcriptional priming of GSDMD-mediated pyroptosis confers the potent immune response to *Mll4*⁻/⁻ tumors

Given the potent immunostimulatory effects of GSDMD-mediated pyroptosis, we, therefore speculated that transcriptional induction of

**Fig. 3 | *Mll3* or *Mll4* ablation increases the cytosolic dsRNAs level to elicit interferon response through attenuation of RISC function. a** Flowchart for total RNA-seq analyses of indicated B16 tumor cells from C57BL/6J mice (*n* = 2). **b, k** M-versus-A (MA) and volcano plots showing differentially expressed protein-coding (**b**) and ERV (**k**) transcripts in *Mll4⁻/⁻* tumor cells (*n* = 2). **c** Gene set enrichment analysis (GSEA) of differentially expressed genes in *Mll4⁻/⁻* tumor cells (*n* = 2). **d, e, h** Heatmaps showing Z-score expression of indicated genes in control and *Mll4⁻/⁻* tumor cells (*n* = 2). **f, g** Cell-surface levels of free (**f**) and antigen-bound MHC I (**g**) in indicated B16-Ova cells (mean ± SEM, *n* = 3). **i, j** Tumor growth (**i**) and survival curves (**j**) of C57BL/6J mice bearing indicated B16 tumors (mean ± SEM, *n* = 5). **l** Representative images and quantification of intracellular dsRNA immunostaining in control and *Mll4⁻/⁻* B16 cells with or without RNase III treatment (mean ± SEM, *n* = 50). Scale bar, 10 µm. **m** Schematic diagram of dsRNAs regulation by RISC complex. **n** RNA-seq FPKM value for indicated genes in control and *Mll4⁻/⁻* tumor

cells (mean ± SEM, *n* = 2). **o, p** Spearman's correlation for the expression of *MLL4* with *AGO2* in all TCGA tumor types (**o**, left), SKCM (**o**, right), and CCLE cancer cell lines with or without tissue origin categorization (**p**) Red dots indicate cancer types with significant correlations. Blue line, linear regression fit; Shaded area, 95% confidence interval. Value in (**p**, left) indicates the number of cancer cell lines of same tissue origin. **q–s** TCGA pan-cancer analyses of the Spearman's correlation and for indicated pairs. **t** Growth curves for the indicated B16 tumors in C57BL/6J mice were shown as mean ± SEM (*n* = 5). **u, v** Frequency (**u**) and cytotoxicity (**v**) of CD8⁺ T cells in tumors described in (**t**). Quantifications were shown as mean ± SEM (**u**, B16 Vector and B16 Ago2, *n* = 5; Mll4KO Vector and Mll4KO Ago2, *n* = 6. **v** *n* = 5). Statistical significance was determined by quasi-likelihood F test (**b, k**), permutation test (**c**), two-tailed unpaired t (**f, g, l, u, v**), two-way ANOVA (**i, t**), cor.test (**o–s**) or log-rank (Mantel−Cox) test (**j**). *\*P < 0.05; \*\*P < 0.01; \*\*\*P < 0.001.*

*Gsdmd* may have an important function in tumor-cell pyroptosis induction and immune elimination of *Mll4⁻/⁻* tumors. To test this hypothesis, we depleted GSDMD expression in *Mll4⁻/⁻* B16-Ova cells and co-cultured them with OT-I CD8⁺ T cells (Fig. 5a, b, and Supplementary Fig. 5a-c). GSDMD depletion significantly reduces the release of LDH and the preloaded calcein AM, and abrogates *Mll4⁻/⁻* B16-Ova cell death elicited by antigen-specific CD8⁺ T cells in vitro (Fig. 5b, c and Supplementary Fig. 5d). In addition, GSDMD depletion also impairs extracellular release of HMGB1 from *Mll4⁻/⁻* B16-Ova cells, and impedes the activation and cytotoxic activity of antigen-specific CD8⁺ T cells in vitro (Fig. 5d and Supplementary Fig. 5e, f), indicating elevated GSDMD is indispensable for pyroptosis induction and cytotoxic killing of *Mll4⁻/⁻* B16-Ova cells by CD8⁺ T cells in vitro.

We next examined whether GSDMD induction mediates the immunosuppression of *Mll4⁻/⁻* tumors in vivo. GSDMD depletion partially rescues the growth defects of *Mll4⁻/⁻* B16 tumor cells in immune-competent mice and mitigates the survival benefits conferred by *Mll4* ablation (Fig. 5e, f). Flow cytometry analyses of the TME revealed that GSDMD depletion impairs the infiltration of total and CD8⁺ T lymphocytes, as well as attenuates the activation and cytotoxic function of intratumoral CD8⁺ T cells in *Mll4⁻/⁻* melanomas (Fig. 5g–j), indicating transcriptional induction of GSDMD-mediated pyroptosis augments the anti-tumor immune response to suppress *Mll4⁻/⁻* B16 tumor progression in immune-competent mice.

### *Mll4* ablation indirectly derepresses *Gsdmd* and inflammatory caspases by attenuating the expression of DNA methyl-transferases through enhancer decommissioning
MLL4 is an established transcriptional co-activator and its target genes are unlikely to be directly upregulated upon its depletion[32,36,55,56]. We thus reasoned that transcriptional induction of *Gsdmd* and inflammatory caspases by *Mll4* loss may be partially mediated through the downregulation of transcriptional suppressors that act as direct targets of MLL4. Pyroptotic executors generally function as tumor suppressors and some of them have been reported to be epigenetically silenced in many cancers through promoter DNA hypermethylation[40,57–59]. GSEA analysis found prominent upregulation of DNA demethylation activity in *Mll4⁻/⁻* tumor cells (Fig. 3c), leading us to explore whether alternation of DNA methylation underlies the transcriptional reactivation of pyroptotic pathway by *Mll4* depletion. *Mll4* deletion leads to a noticeable reduction in the mRNA and protein levels of *Dnmt3a* and *Dnmt1* and a mild increase of *Tet2* expression in sorted B16 cells and bulk tumors (Fig. 6a, b and Supplementary Fig. 6a, b). Analyses of RNA-seq datasets in TCGA and CCLE for human tumors and cancer cell lines revealed prominent positive correlations for *MLL4* expression and the mRNA levels of *DNMT3A* or *DNMT1* (Fig. 6c and Supplementary Fig. 6c), indicating MLL4 may have a universal role in transcriptional activation of DNA methyltransferases in human cancer cells of various tissue origins. Knockdown of *MLL4* leads to a marked decrease in DNMT3A and

DNMT1 levels in several human cancer lines we examined (Supplementary Fig. 6 d).

We and others have previously shown that MLL4 boosts the expression of its target genes through enhancer occupancy and regulation[27,32,36]. H3K27ac, together with H3K4me1, marks active enhancers that are transactivating their target gene expression[26]. In comparison to most active enhancers (called typical enhancers), a small set of active enhancers are decorated with exceptionally high levels of H3K27ac and H3K4me1 and usually cluster together to form large genomic domains, which are termed super enhancers. Super enhancers bear increased activity to drive transcriptional programs for establishing and maintaining cellular identity[60,61]. To characterize the impact of *Mll4* ablation on the epigenomic landscape and transcriptional strength of enhancers, we conducted ChIP-seq for H3K4me1 and H3K27ac in control and *Mll4⁻/⁻* B16 cells. *Mll4* ablation results in a widespread reduction of H3K4me1 and H3K27ac signals at both typical and super enhancers and significantly decreases the expression of their associated genes (Fig. 6d–h). Intriguingly, a preferential reduction of expression was observed for super-enhancer-associated genes as compared to genes associated with typical enhancers (Fig. 6i), suggesting *Mll4* ablation leads to more severe defects of super-enhancer activity compared to typical enhancers. Inspection of individual enhancers of genes downregulated by *Mll4* ablation revealed that *Dnmt1* and *Dnmt3a* are linked to typical enhancers while *Ago2* is associated with a super-enhancer in wild-type B16 cells (Fig. 6j). *Mll4* deletion causes pronounced reduction of H3K27ac signal at enhancers of both *Dnmt3a* and *Dnmt1* genes and converts the super-enhancer of *Ago2* into a typical enhancer (Fig. 6j, k, and Supplementary Fig. 6e), suggesting MLL4 loss decreases expression of its target genes through impairing their enhancer activity.

DNMT3A and DNMT1 silence gene expression via implementing and/or maintaining promoter DNA methylation of their target genes[62,63]. To determine whether decreased expression of *Dnmt3a* and *Dnmt1* mediates the effects of *Mll4* ablation on gene depression, we first assessed their levels at *Gsdmd*, *Casp1*, and *Casp11* loci in B16 cells. Quantitative ChIP PCR (ChIP-qPCR) analyses revealed notable binding of both DNMT3A and DNMT1 near the transcription start sites (TSSs) of these genes (Fig. 6l). Attenuating expression of *Dnmt3a* and *Dnmt1* or pharmacologically inhibiting their DNA methyltransferase activity by Decitabine leads to a marked elevation of *Gsdmd*, *Casp1*, and *Casp11* expression at both transcripts and protein levels in in vitro cultured B16 cells (Fig. 6m and Supplementary Fig. 6f, g). In addition, Decitabine treatment impairs tumor progression in immunocompetent mice and gives rise to a marked increase of both full-length and the cleaved N-terminal fragment of GSDMD in bulk melanomas in vivo (Supplementary Fig. 6h–k), further substantiating a key role of DNMT3A and DNMT1 in transcriptional silencing of GSDMD-mediated pyroptotic pathway in melanoma cells. In line with a reduction of DNMT3A and DNMT1 bulk levels, *Mll4* loss leads to a dramatic decrease in DNMT3A

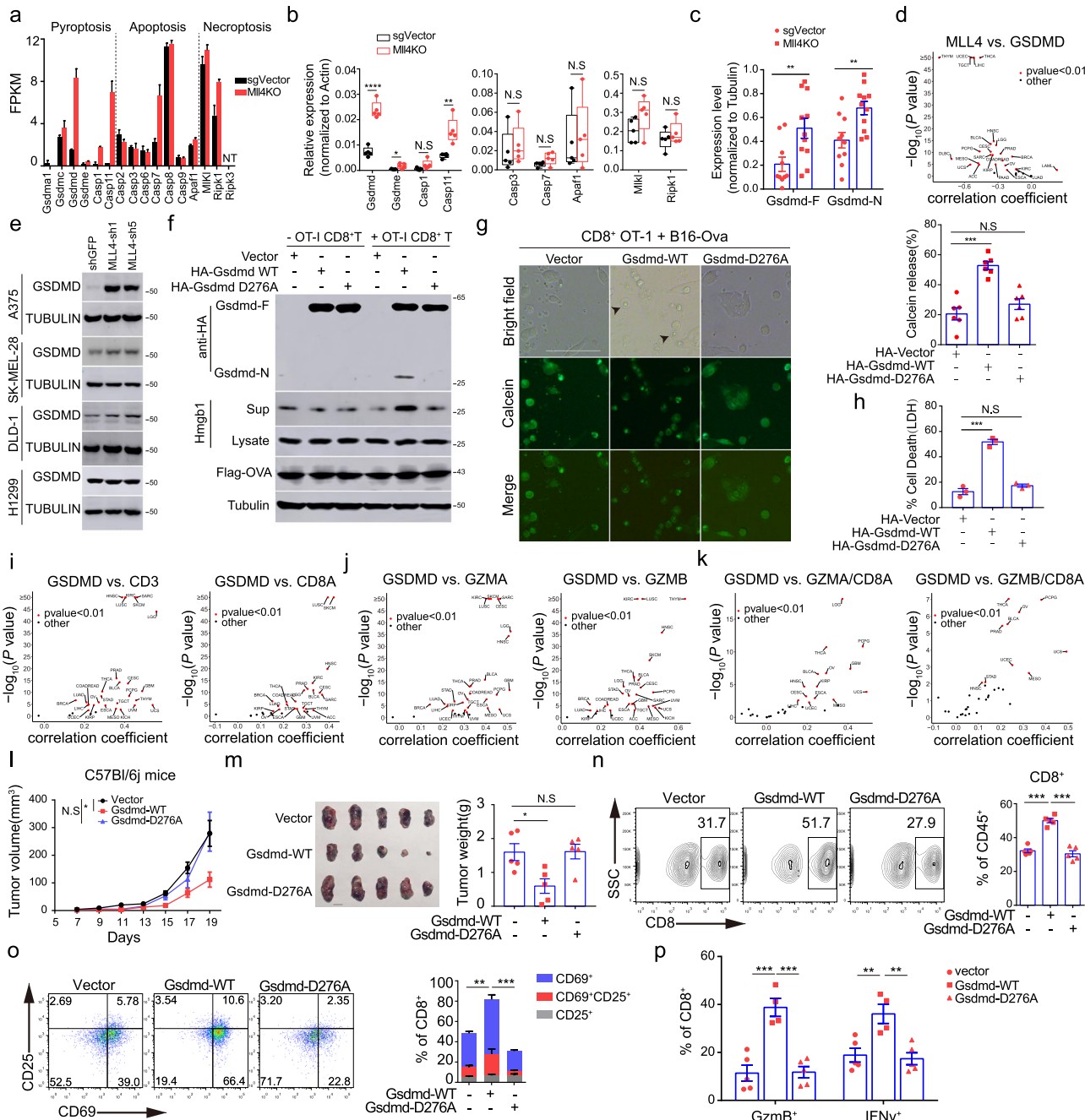

**Fig. 4 | Mll4 ablation derepresses GSDMD to mediate pyroptosis in B16 melanoma cells. a** RNA-seq FPKM value for indicated genes in control and $Mll4^{-/-}$ tumor cells (mean ± SEM, $n = 2$). **b** Box plot showing expression of indicated genes in control and $Mll4^{-/-}$ B16 bulk tumors ($n = 5$). Center line, mean; box bounds, upper and lower quartile; whisker, maximal and minimal value ($n = 5$). **c** Levels of full-length or the truncated N-terminal GSDMD protein in control and $Mll4^{-/-}$ B16 bulk tumors are shown as mean ± SEM (sgVector, $n = 10$; Mll4KO, $n = 11$). **d, i–k** Volcano plots showing TCGA pan-cancer RNA-seq analyses of the Spearman's correlation for the indicatedcomparisons. **e** Immunoblotting analyses of GSDMD level in human cancer cell lines with or without $MLL4$ knockdown. **f, h** OT-1 T cells were incubated with B16-Ova cells expressing vehicle, WT or mutant $Gsdmd$ at an E/T ratio of 10:1 for 24 h. Levels of indicated protein in co-culture supernatant and tumor-cell extracts were analyzed by immunoblotting (**f**). Tumor-cell death was determined by LDH release and shown as mean ± SEM from technical triplicates in one of

biological replicates (**h**). **g** B16-Ova cells expressing vehicle, WT or mutant $Gsdmd$ were preloaded with calcein AM and then co-cultured with OT-I CD8+ T cells at an E/T ratio of 10:1 for 12 h. Representative images are shown with arrowheads indicating tumor cells undergoing pyroptosis. Scale bar: 20 μM. Quantification is shown as mean ± SEM from three biological replicates. **l, m** Tumor growth over time (**l**), photographs and weight quantification at the time of experimental endpoint (**m**) in C57BL/6J mice implanted with indicated B16-Ova cells were shown as mean ± SEM ($n = 5$). **n–p** Infiltration (**n**), activation (**o**) and cytotoxicity (**p**) of CD8+ T cells in tumors of C57BL/6J mice implanted with indicated B16-Ova cells. Quantifications were shown as mean ± SEM (vector and Gsdmd-D276A, $n = 5$; Gsdmd-WT, $n = 4$). All immunoblots are representative from biological replicates. Statistical significance was determined by two-tailed unpaired $t$ test (**b, c, g, h, m, n, o, p**), cor.test (**d, i–k**) or two-way ANOVA (**l**). *$P < 0.05$; **$P < 0.01$; ***$P < 0.001$.

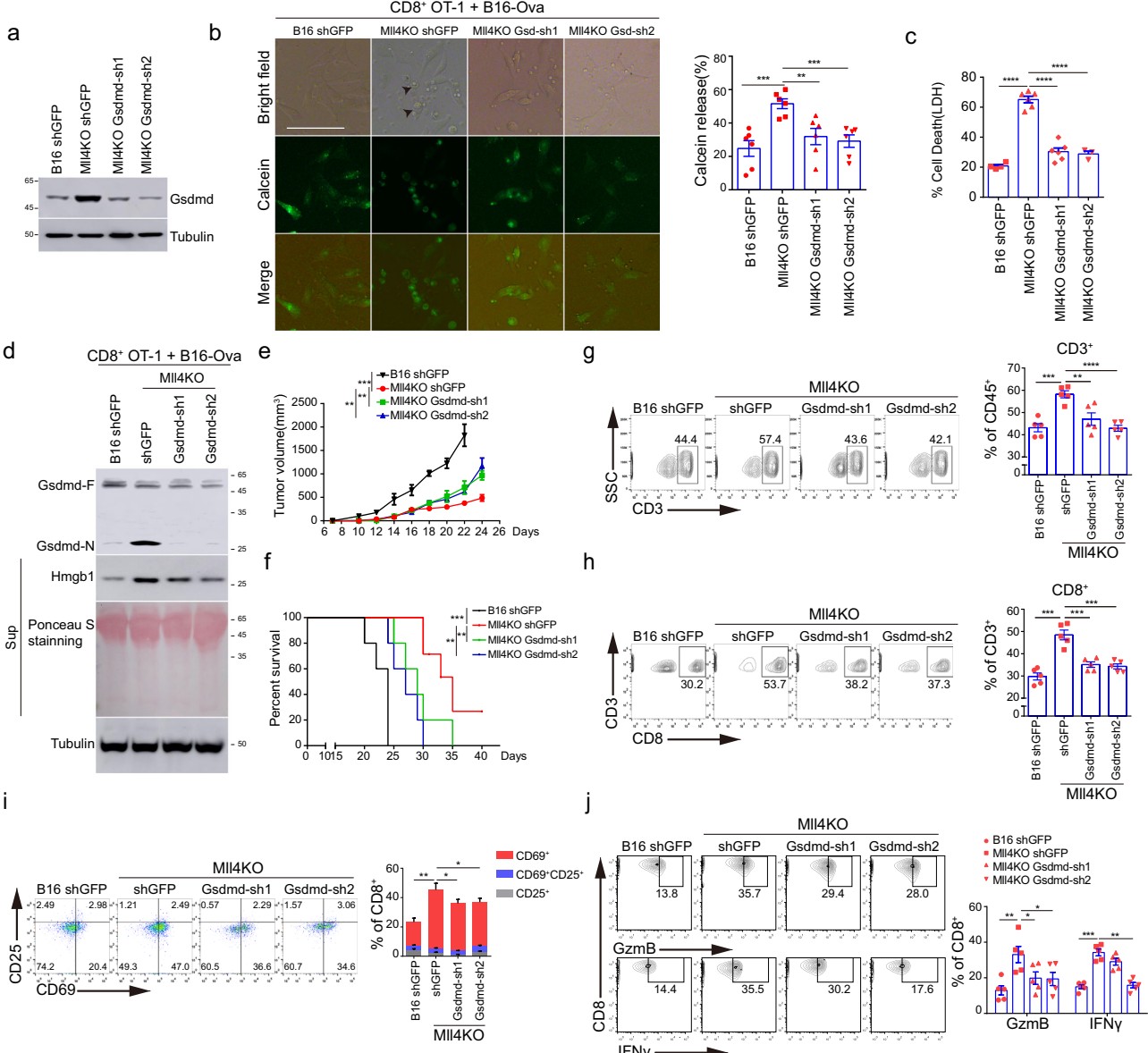

**Fig. 5 | GSDMD depletion attenuates tumor-cell pyroptosis and the immuno-suppressive effects of CD8+ T cells on *Mll4*⁻/⁻ tumors.** **a** GSDMD level in control or *Mll4*⁻/⁻ B16-Ova cells lentivirally transduced with *Gsdmd* shRNAs. Shown are representative results from two independent experiments. **b** Control or *Mll4*⁻/⁻ B16-Ova cells depleted for GSDMD expression were preloaded with calcein AM and then co-cultured with OT-I CD8+ T cells at an E/T ratio of 10:1 for 12 h. Representative images are shown with arrowheads indicating tumor cells undergoing pyroptosis. Scale bar: 20 μm. Quantification of released calcein AM is shown as mean ± SEM from five technical replicates in one of two independent experiments **c** Control or *Mll4*⁻/⁻ B16-Ova cells depleted for GSDMD expression were treated as in (**b**) except for 24 h of co-culture. T-cell-mediated killing was determined by LDH release-based cell death analysis and shown as mean ± SEM from technical triplicates in one

experiment of biological replicates. **d** Control or *Mll4*⁻/⁻ B16-Ova cells depleted for GSDMD expression were treated as in (**b**) except for 24 h of co-culture. GSDMD cleavage and HMGB1 release from target cells were analyzed by immunoblotting. Shown are representative results from one experiment of biological replicates. **e**, **f** Control or *Mll4*⁻/⁻ B16 cells depleted for GSDMD expression were inoculated into C57BL/6J mice. Tumor growth (**e**) and Kaplan–Meier curves for mice survival (**f**) were recorded (mean ± SEM, *n* = 5). **g–j** Control or *Mll4*⁻/⁻ B16 cells depleted for GSDMD expression were inoculated into C57BL/6J mice. Intratumoral frequency of total T (**g**) and CD8+ T (**h**) cells, activation (**i**) and cytotoxicity (**j**) of infiltrated CD8+ T cells were analyzed by flow cytometry and represented as mean ± SEM (*n* = 5). Statistical significance was determined by two-tailed unpaired t (**b**, **c**, **g–j**), two-way ANOVA (**e**) or log-rank (Mantel–Cox) test (**f**). *P < 0.05; **P < 0.01; ***P < 0.001.

and DNMT1 occupancy and the promoter DNA CpG methylation at *Gsdmd*, *Casp1*, and *Casp11* loci (Fig. 6n, o). Ectopic expression of *Dnmt3a* and *Dnmt1 alone* or in combination abolishes the increased expression of *Gsdmd* in *Mll4*⁻/⁻ B16 cells (Fig. 6p), suggesting defects in DNMT3A and DNMT1 -catalyzed promoter CpG methylation are responsible for the transcriptional derepression of GSDMD-triggered pyroptosis in *Mll4*⁻/⁻ melanomas. Collectively, these data indicate that *Mll4* ablation impairs the enhancer activity and decreases the expression of DNA methyltransferases to predispose tumor cells to GSDMD-mediated pyroptosis.

## GSDMD-mediated pyroptosis confers the immunotherapeutic efficacy of anti-PD-1 blockade in *Mll4*⁻/⁻ tumors

As GSDMD-mediated pyroptosis promotes tumor infiltration and activation of CD8+ T lymphocytes (Fig. 4), we, therefore, speculated that GSDMD-dependent pyroptosis will synergize with immune checkpoint blockage therapies to potentiate anti-tumor immunity. To test this hypothesis, C57BL/6J mice were implanted subcutaneously with B16-Ova cells that are reconstituted with the vehicle, wild-type, or D276A mutant GSDMD and then administered intraperitoneally with an anti-PD-1 antibody or isotype IgG control (Fig. 7a). PD-1 blockade

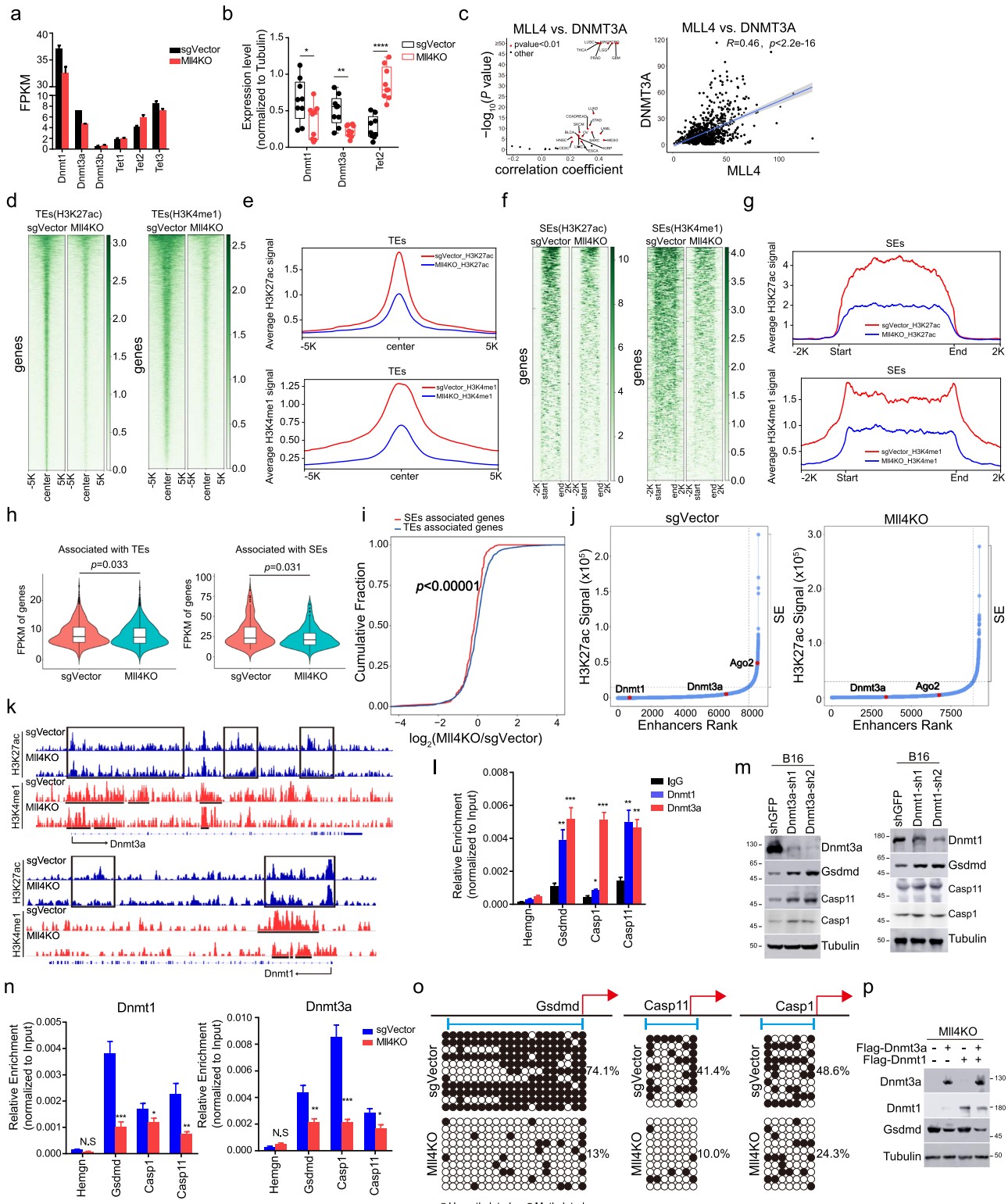

alone has no visible effects on wild-type B16 tumor growth, which is in line with earlier reports showing B16 tumor cells are of poor immunogenicity and refractory to PD-1-PD-L1 checkpoint blockade (Fig. 7b)[64,65]. Although wild-type *Gsdmd* expression attenuates tumor growth in immune-competent mice as noted before, anti-PD-1 administration further suppresses tumor progression and leads to a nearly invisible tumor burden at the experimental endpoints, which was not observed for D276A GSDMD-reconstituted tumors (Fig. 7b, c). Moreover, anti-PD-1 treatment further increases the abundance and

promotes the cytotoxic potential of CD8⁺ T cells in tumors ectopically expressing wild-type GSDMD (Fig. 7d, e), suggesting PD-1 blockade and GSDMD can synergize with each other to potentiate anti-tumor efficiency of cytotoxic T lymphocytes.

To determine the translational significance of the above findings, we first analyzed the relationship between the expression of a pyroptotic gene signature in tumors and the clinical response to immune checkpoint blockade in one cohort of metastatic melanoma patients treated with an anti-CTLA4 blocking antibody. Higher levels of the

**Fig. 6 | *Mll4* deletion transcriptionally elevates *Gsdmd* expression through attenuating enhancer activity of *Dnmt3a* and *Dnmt1*. a** RNA-seq FPKM value for indicated genes in sorted control and *Mll4*⁻/⁻ tumor cells (mean ± SEM, *n* = 2). **b** Levels of indicated protein in control or *Mll4*⁻/⁻ B16 bulk tumors (mean ± SEM, *n* = 9). **c** Volcano and scatter plots showing the Spearman's correlation between *MLL4* and *DNMT3A* mRNA levels across TCGA human cancer types (left) and CCLE cancer cell lines (right). Shaded area, 95% confidence interval. **d–g** Heatmaps and metaplots showing H3K27ac and H3K4me1 signals at typical enhancers (**d, e**) and super enhancers (**f, g**) in control and *Mll4*⁻/⁻ B16 cells. **h** Violin plots comparing transcript levels of typical or super enhancers associated genes in sorted control and *Mll4*⁻/⁻ B16 tumor cells (*n* = 2). Center line, median; box, upper and lower quantiles; whisker, maximal and minimal values. **i** Cumulative distribution plot of log2 fold changes in RNA expression of genes associated with typical and super enhancers in sorted control and *Mll4*⁻/⁻ B16 tumor cells (*n* = 2). **j** H3K27ac signal in an increasing order across all enhancers in control and *Mll4*⁻/⁻ B16 cells. Super enhancers are defined as the group of enhancers above the inflection point of the curve. **k** IGV browser tracks showing H3K4me1 and H3K27ac occupancy at indicated loci in control and *Mll4*⁻/⁻ B16 cells. **l, n** DNMT3A and DNMT1 occupancy at indicated genomic loci in B16 cells (**l**) or in control and *Mll4*⁻/⁻ B16 cells (**n**) were shown as mean ± SD from technical triplicates in one of two biological replicates. **m, p** Immunoblotting analyses of indicated protein in B16 cells with *Dnmt3a* or *Dnmt1* depletion (**m**) or in *Mll4*⁻/⁻ B16 cells expressing *Dnmt3a* and *Dnmt3b* alone, or in combination (**p**) Shown are representative results from biological replicates. **o** Lollipop representation of DNA methylation status at indicated gene promoter. Filled circles, methylated CpGs; open circles, unmethylated CpGs. Statistical significance was determined by two-tailed unpaired t (**b, l, n**), cor.test (**c**), two-sided Mann–Whitney *U* test (**h**) or two-sample Kolmogorov–Smirnov test (**i**). n.s not significant, *P < 0.05; **P < 0.01; ***P < 0.001.

pyroptotic gene signature in tumor samples are associated with durable clinical benefits in patients treated with anti-CTLA4 therapy[66] (Fig. 7f). We next sought to determine if the pyroptotic gene signature is similarly correlated with the responsiveness of tumors to PD-1-PD-L1 checkpoint blockade in cohorts of melanoma patients or patients bearing othercancer types. Single-sample GSEA using pyroptotic gene Signature score (S-score) revealed that the majority of responding tumors tend to have a positive S score while most non-responsive tumors, if not all, are negative for S-score in several cohorts of melanoma patients and one cohort of patients with gastric cancer[67–71] (Fig. 7g and Supplementary Fig. 7a, c, d), indicating that tumors with high expression of pyroptotic gene signature correlate with a stronger response to PD-1-PD-L1 checkpoint blockade therapies. Consequently, greater overall- and progression-free survival benefits were observed in patients with metastatic melanoma or gastric cancer expressing higher levels of pyroptotic gene signature[68,69,71] (Fig. 7h and Supplementary Fig. 7b, e).

Though *Mll4* deletion decreases tumor progression in immune-competent mice in general, our transcriptomic and flow cytometric analyses also revealed a pronounced increase of *Pd-l1* expression in *Mll4*⁻/⁻ melanoma cells in vivo (Fig. 7i, j), suggesting that loss of *Mll4* could also attenuate cytotoxic potential and thus compromise the anti-tumor effects of infiltrated CD8⁺ T cells to some degree. These findings lead us to propose a synergistic and cooperative effect for PD-1-PD-L1 blockade and *Mll4* ablation in promoting tumor immunity and the immune clearance of melanoma cells in vivo (Fig. 7k). Indeed, we found that anti-PD-1 treatment further suppresses the *Mll4*⁻/⁻ melanoma growth and reduce the tumor burden to nearly invisible size in immune-competent mice at the endpoint of the experimental schedule (Fig. 7l, m), proving the synergistic effects of the anti-PD-1 blockade and *Mll4* loss in immunosuppression of tumor progression. To investigate whether or not GSDMD-mediated pyroptosis mediates the tumor-suppressive effects of *Mll4* deletion or combination of *Mll4* deletion and anti-PD-1 blockade in vivo, we administered mice with disulfiram to specifically block GSDMD-mediated pyroptosis through inhibiting the pore formation of the cleaved N-terminal GSDMD fragment on plasma membrane[72] (Fig. 7k). Blocking GSDMD-mediated pyroptosis partially rescues the growth defects of *Mll4*⁻/⁻ melanoma cells and markedly increases the tumor burden in immune-competent mice treated with or without PD-1 blockade antibody (Fig. 7l, m). Taken together, these results demonstrate that GSDMD-mediated pyroptosis promotes tumor immunity and can sensitize refractory tumors to cytotoxic killing by tumor-infiltrated T lymphocytes, which at least partly account for the drastic suppression of *Mll4*⁻/⁻ tumor progression in immune-competent mice treated with or without anti-PD-1 immunotherapy.

## Discussion

Deregulation of enhancers through alternations in their DNA sequence and/or associated chromatin regulators has been recognized as a primary mechanism for the aggressive biology of cancer cells. However, the role of tumor-cell enhancer regulation in cancer immunity and immunotherapeutic response is poorly understood. Starting from the bioinformatic analysis of publicly available CRISPR/Cas9 datasets for screening tumor-cell-intrinsic regulators of tumor immunity, we identified and characterized a previously unappreciated role of tumor-cell enhancer regulation by MLL3 and MLL4 in the control of T-cell-mediated killing and tumor immune response in multiple syngeneic murine cancer models. Ablation of *Mll3* or *Mll4* elicits overt interferon response and tumor-cell pyroptosis in the context of an intact host immune system, which at least partially results from increased innate sensing of cytosolic dsRNAs and transcriptional priming of components in the GSDMD-mediated pyroptotic pathway, respectively. Mechanistic studies with epigenetic and transcriptomic profiling showed that MLL4 promotes activation of both typical and super enhancers to increase the expression of their nearby genes, including the RISC component *Ago2* and the DNA methyltransferases *Dnmt1* and *Dnmt3a* in tumor cells. Attenuation of *Ago2* expression partially accounts for the increased tumor immunogenicity and the augmented immune response to *Mll4*⁻/⁻ tumor cells, probably through interfering with dsRNA homeostasis and eliciting interferon responses. In addition, our results indicated that decreased *Dnmt1* and *Dnmt3a* expression abrogates promoter CpG methylation and depresses inflammatory *Casp1* and *Casp11* as well as pyroptotic executor *Gsdmd* to transcriptionally prime *Mll4*⁻/⁻ tumor cells for pyroptosis. Our data further indicated that GSDMD-mediated pyroptotic cell death is immunostimulatory and is indispensable for the increased spontaneous anti-tumor immune response and the enhanced sensitivity to anti-PD-1 blockade in *Mll4*⁻/⁻ melanomas (Fig. 7n).

Consistent with the loss-of-function mutations identified throughout both genes in various human cancer types, the tumor-suppressive function of MLL3 and MLL4 has been well established using genetic models of multiple murine cancers[34,35,55]. Nevertheless, our study pinpoints an essential role for MLL3 and MLL4 in suppression of tumor immunity and anti-tumor immune response in diverse syngeneic murine tumor models, thus promoting tumor formation and development. In line with our mouse work, analysis of TCGA human cancer datasets revealed that expression of *MLL3* or *MLL4* is inversely correlated with the levels of markers for T-cell infiltration and function. One prominent question emerges is how tumors develop and progress in light of genetic inactivation of factors, like MLL3 and MLL4, with both tumor suppressive and promoting function. It is now well recognized that tumor develops through competition between their aggressive features and the immune suppression of the surrounding tumor microenvironment[14]. We speculated that some of the epigenetic factors may possess essential roles in the regulation of both tumor aggressive behaviors and tumor immunity, and thus can favor both tumor development and suppression. In accordance with our assumption, prior studies have found conditional deletion of *Mll4* confers growth advantages and causes blockade of differentiation

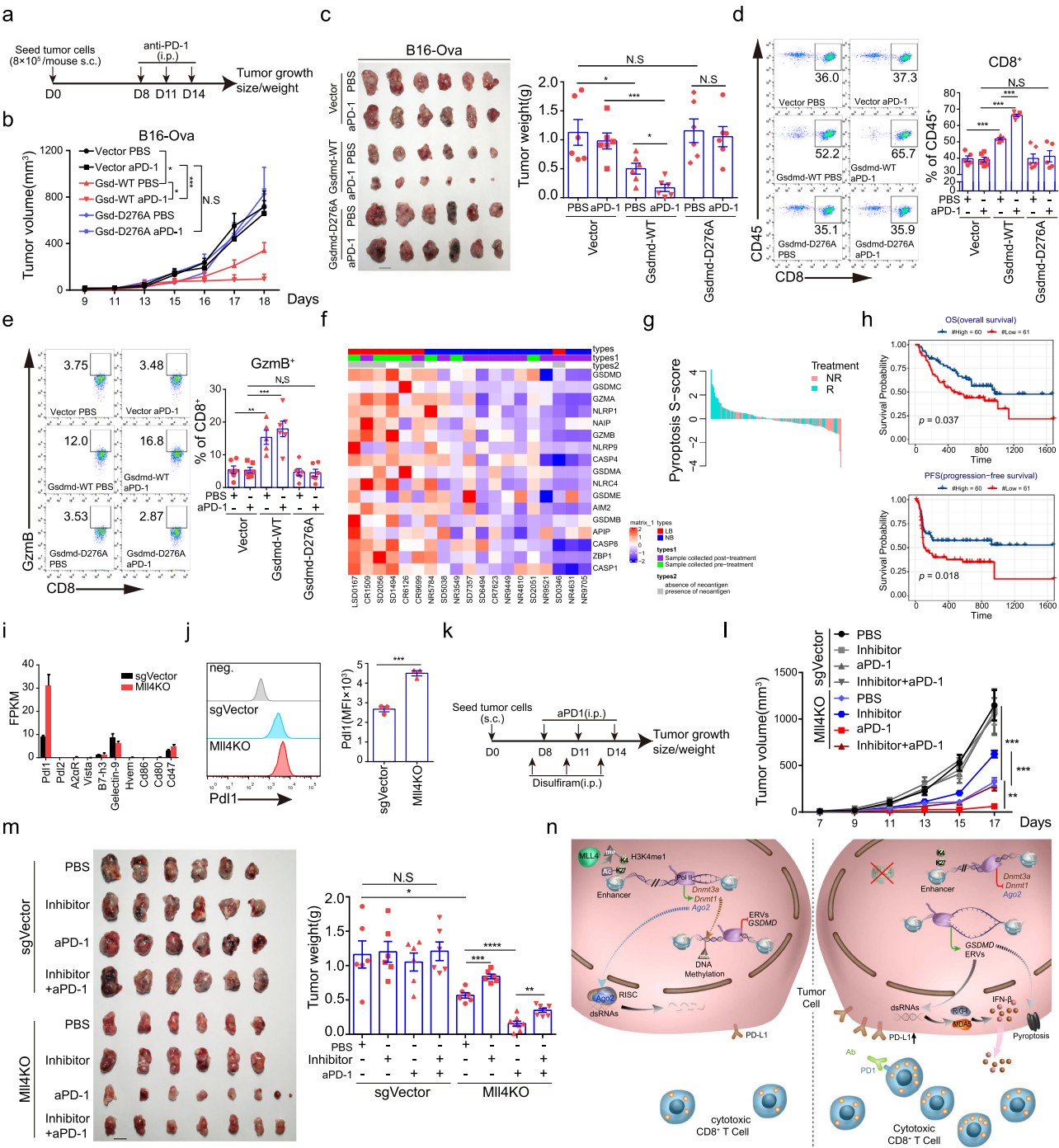

**Fig. 7 | Blocking GSDMD-mediated tumor-cell pyroptosis abrogates immunotherapeutic efficacy of anti-PD-1 blockade in *Mll4*[-/-] tumors. a** Experimental schedule for anti-PD-1 treatment in C57BL/6J mice. **b** Tumor growth curve in mice described in (**a**) (mean ± SEM, *n* = 6). **c** Tumor images and weight quantification at the time of experimental endpoint in mice described in (**a**) (mean ± SEM, *n* = 6). **d, e** Frequency (**d**) and cytotoxicity of CD8[+] T cells (**e**) in tumors of mice described in (**a**) (mean ± SEM, *n* = 6 for all groups except for Gsdmd-WT treated with PBS, in which n equals 5_). **f** Heatmap showing expression of pyroptosis-related genes in tumors from anti-CTLA4 treated metastatic melanoma patients who achieve long-term (LB, complete and partial response or progression-free survival over 6 months) or no benefit (NB, progressive disease)[66]. **g** A waterfall plot showing the pyroptosis S-score in tumors from metastatic melanoma patients who respond or do not respond to anti-PD-1 immunotherapy[68]. **h** Kaplan–Meier curves showing overall and progression-free survival of melanoma patients splitted by pyroptosis S-score[68] (high S-score, *n* = 60; low S-score, *n* = 61). **i** RNA-seq FPKM values for negative regulators of anti-tumor immune response in sorted control or *Mll4*[-/-] B16 tumor cells of C57BL/6 J mice (mean ± SEM, *n* = 2). **j** Representative flow histograms and quantification of cell-surface expression of PD-L1 in control and *Mll4*[-/-] B16 cells. Quantification of MFI was shown as mean ± SEM from technical triplicates in one of biological replicates. **k** Experimental design for the treatment of C57BL/6J mice with control or *Mll4*[-/-] B16 tumors. **l, m** Tumor growth (**l**), and tumor images and weight quantification (**m**) were shown as mean ± SEM (*n* = 6 for all groups except for Mll4KO groups treated with aPD-1 alone or together with disulfiram, in which n equals 8). **n** A proposed model summarizing the role and molecular mechanisms of Mll4 in immunosuppression and tumor immune evasion. Statistical significance was determined by two-tailed unpaired t (**c, d, e, j, m**), two-way ANOVA (**b, l**) or log-rank test (**h**). *$P < 0.05$; **$P < 0.01$; ***$P < 0.001$.

program and ferroptosis suppression, thus synergizing with genetic lesions of additional genes to promote tumor development in multiple mouse tissues[34,35,55,73,74]. We and others found that *Mll4* ablation elevates the expression of multiple chemokines, including CXCL9 and CXCL10, and the increased levels of molecules responsible for antigen processing and presentation[75], which augments CD8[+] T-cell-mediated anti-tumor immune response to suppress tumor development[16,49]. Concurrent with these changes, we also found a notable increase of PD-L1 expression in *Mll4*[−/−] tumor cells, which may compromise the tumor-cell-killing efficacy of intratumoral cytotoxic T-cell[76]. Indeed, immunotherapeutic treatment blocking the PD-1-PD-L1 axis has a striking synergistic effect with *Mll4* ablation on tumor suppression in immune-competent mice, indicating *Mll4* loss could sensitize the refractory tumor cells to anti-PD-1 blockade treatment. Given the frequent loss of function mutations and epigenetic silencing of *MLL3* and *MLL4* in diverse cancer types[33,73], our findings suggest that the genetic and expression status of *MLL3* and *MLL4* may serve as biomarkers for cancer patient stratification in immune checkpoint blockade therapies such as anti-PD-1 or anti-PD-L1 treatment and in predicting immunotherapeutic efficacy and clinical prognosis.

Transcriptomic profilings reveal a marked increase in the expression of multiple cytosolic sensors for dsRNAs and dsDNAs as well as elevated levels of many ERVs in *Mll3*[−/−] or *Mll4*[−/−] melanoma cells. As mobile and major constituents of the mammalian genome, transcription of the repetitive ERV elements is normally silenced by epigenetic mechanisms to prevent their propagation and to preserve genome integrity[52,77]. Recent studies reported that targeting epigenetic mechanisms in cancer cells, including DNA methylation by DNMTs, histone methylation, and demethylation by SETDB1, LSD1, and KDM5B could derepress ERVs transcription and lead to induction of interferon response[19–22]. As an epigenetic co-activator for gene transcription, our data reveal that loss of *Mll4* abrogates enhancers to downregulate *Dnmt3a* and *Dnmt1* expression in murine tumor cells, suggesting the increased ERVs transcripts we noted in *Mll4*[−/−] melanoma cells may indirectly result from the defects in DNA methylation-mediated epigenetic silencing.

It has been documented that transcripts derived from bidirectional ERVs transcription could form dsRNAs to trigger interferon response[78]. dsRNAs can further be resolved by RISC components to prevent dsRNA stress from triggering interferon pathway activation[53,79]. We found that *AGO2* transcript is positively correlated with *MLL4* expression in the vast majority of human TCGA cancer types and CCLE cancer cell lines and loss of MLL4 decreases the expression of AGO2 in murine B16 cells and a few human cancer lines, indicating a direct and general role that MLL4 plays in transcriptional suppression of AGO2 expression in cancer cells irrespective of their tissue origin. Through coordinating both ERVs and AGO2 expression, MLL4 ablation results in a dramatic increase of cytosolic dsRNAs and consequently triggers interferon signaling in murine tumor cells. It will be intriguing to determine if there are any differences in the levels of AGO2 and dsRNAs between human tumor cells with and without loss of mutations in MLL4. In addition, future studies should also be warranted to explore if the induced ERVs contribute to the gross antigen pools of *Mll4*[−/−] tumor cells as a few ERV transcripts have been shown to encode short peptides that can be presented by MHC I for specific T-cell activation[23,80,81].

Inflammatory caspases-mediated GSDMD cleavage after residue D276 mediates pyroptotic cell death in response to inflammasome activation[38,39]. Latest studies reported that two other members of the GSDM family, GSDMB and GSDME, are cleaved by granzymes of killer lymphocytes to potentiate anti-tumor immunity via induction of tumor-cell pyroptosis[41,42]. Whether anti-tumor immune response also leads to GSDMD cleavage to elicit pyroptosis in tumor cells is largely unknown. Our results present compelling evidence for the GSDMD cleavage and induction of GSDMD -mediated pyroptosis in murine melanoma cells when attacked by antigen-specific cytotoxic T

lymphocytes. GSDMD -mediated tumor-cell pyroptosis also stimulates anti-tumor immune response and promotes tumor immunosuppression in immunocompetent mice. Moreover, we found expression of pyroptotic pathway components, including *GSDMD*, *CASP1*, and *CASP4* is positively correlated with clinical response to anti-PD-1 or anti-PD-L1 blockade treatment and is predictive of clinical prognosis in multiple cohorts of patients with diverse cancers, suggesting the induced expression of *Gsdmd* and inflammatory caspases confers the increased tumor-suppressive effects of the PD-1 blockade in *Mll4*[−/−] murine melanoma model. Indeed, pharmacologic blocking of GSDMD -mediated pyroptosis abrogates immunotherapeutic sensitivity of *Mll4*[−/−] melanomas to anti-PD-1 treatment. Though little is known about the molecular mechanisms underlying GSDMD cleavage in this scenario, it is clear at present that cleavage occurs at the same site as cut by inflammatory caspases as D276A substitution blocks GSDMD proteolysis and abolishes the stimulating effects of GSDMD on anti-tumor immunity. Future studies are warranted to determine if the induced inflammatory caspases or murine granzymes mediate GSDMD cleavage and pyroptotic induction in *Mll4*[−/−] tumor cells.

Expression of pyroptotic executors varies among normal tissues and is often silenced with the repressive chromatin modifications, such as DNA methylation, during malignant transformation[40,57]. In line with the antagonistic relationship between DNA methylation and gene activation, our DNA methylation analysis revealed a dramatic decrease of CpG methylation levels at promoter regions of *Gsdmd*, *Casp1*, and *Casp11* in *Mll4*[−/−] tumor cells. *MLL4* expression is positively correlated with levels of *DNMT1* and *DNMT3A* in most TCGA human cancer types and across human cancer cell lines. Genetic ablation of *Mll4* in murine melanoma cells leads to reduced expression of *Dnm3a* and *Dnmt1* via attenuating their enhancer activities. These results indicated that the effects of MLL4 loss on the levels of GSDMD and its upstream inflammatory caspases are at least in part mediated by decreased expression of DNMT1 and DNMT3A and the resultant DNA methylation reduction at the regulatory regions of these genes. Though we found MLL4 depletion also decreases the levels of DNMT3A and DNMT1 in human cancer cell lines we tested, their transcriptional downregulation may not trigger derepression of GSDMD and other inflammatory caspases in other human cancer cell lines as genetic inactivation of both genes have been observed in hematological malignancy and solid tumors. *Gsdmd* expression is transcriptionally induced by the IRF family transcription factor IRF2 in mouse macrophages[82]. Intriguingly, our RNA-seq data reveals substantial upregulation of IRF2 expression in *Mll4*[−/−] melanoma cells (Fig. 3d). Future work is needed to establish if the induced IRF2 contributes to the depression of *Gsdmd* in *Mll4*[−/−] tumor cells as well.

In addition to pyroptosis, cytotoxic lymphocytes also kill tumor cells through other cell death pathways, such as apoptosis[83]. Intriguingly, previous literature reported that treatment of cells with apoptosis-inducing agents can cause GSDME cleavage and activation by caspase-3 to mediate both pyroptosis in healthy cells and/or secondary necrosis in apoptotic cells in an expression level-dependent manner[57,84]. Due to the intrinsic heterogeneity of tumor cells, the degree of epigenetic derepression of GSDMD after *Mll4* ablation may vary among different single cells, leading to the variable abundance of GSDMD in individual *Mll4*-ablated single cells. Therefore, as in the case of GSDME-mediated pyroptosis and secondary necrosis in cells following apoptotic induction, the variable levels of GSDMD could act similarly to GSDME to mediate both pyroptosis and secondary necrosis to induce the pyroptotic morphology and the increased release of cytosolic contents we observed in *Mll4*-ablated B16 cells when targeted by cytotoxic T lymphocytes.

## Methods

### Cell culture

B16F10 and MC38 are generously provided by H.T at Tsinghua UniversityHEK293T, LLC, and A375 are obtained from American Type

Culture Collection (ATCC). These cell lines were cultured in DMEM supplemented with 10% fetal bovine serum (FBS) and 1% penicillin–streptomycin (PS). DLD-1 and H1299 are from ATCC and maintained in RPMI-1640 plus 10% FBS and 1% PS. SK-MEL-28 cells are from ATCC and cultured in EMEM supplemented with 10% FBS and 1% PS. Primary CD8$^+$ T were cultured in RPMI-1640 supplemented with 10% FBS, 20 mM HEPES, 1 mM sodium pyruvate, 50 µM beta-mercaptoethanol, 10 ng/ml mouse IL-2, 2 mM L-glutamine and 1% PS. Cell lines were authenticated genetically by the providers and we indirectly verified their identity by their morphology, growth behavior or transcriptomic profiles. All cell lines were regularly tested for mycoplasma contamination using the MycoBlue Mycoplasma Detector (Vazyme, D101-01) and maintained at 5% $CO_2$ and 37 °C.

## Mice

6- to 8-week-old male or female C57BL/6J mice and BALB/c nude mice were from the Jackson Laboratory, and B6/JGpt-$Rag1^{em1Cd}$/Gpt ($Rag1^{-/-}$) mice were purchased from GemPharmatech company. C57BL/6J-Tg[TcraTcrb]1100Mjb/J (OT-I TCR transgenic mice) were kindly provided by J. Yang lab (Tianjin Medical University, Tianjin, China). Mice were used in accordance with the protocols (TMUaMEC 2020001) approved by the Institutional Animal Care and Use Committee at the Tianjin Medical University. All mice have free access to food and water and were housed in a pathogen-free environment with a 12:12 dark/light cycle and controlled temperature ($23 \pm 2$ °C) and humidity ($60 \pm 10$%).

## Mouse tumor models

For each mouse, $5 \times 10^5$ tumor cells were injected subcutaneously. Tumor volume was recorded every two days using a caliper. For PD-1 antibody treatments, mice were injected intraperitoneally with 100 µg PD-1 antibody (clone 29 F.1A12) or IgG at day 8, 11, and 14 after tumor-cell inoculation. For inhibitor treatment, when tumor volume reached 100 mm$^3$, tumor-bearing mice were randomly divided into different treatment groups and injected intraperitoneally with 2.5 mg/kg Decitabine or equal volume of saline on day 7,9 and 11 post tumor-cell inoculation. For blocking Gsdmd-mediated pyroptosis in vivo, 50 mg/kg Disulfiram or equal volume of PBS were injected intraperitoneally into mice on day 7, 10, 13. Mice were euthanized by cervical dislocation before tumor size reaches the maximal allowable diameter of 20 mm by the Institutional Animal Care and Use Committee at the Tianjin Medical University.

## Adoptive transfer experiments

For each $Rag1^{-/-}$ mouse, $5 \times 10^5$ Flag-Ova-B16F10 cells were injected subcutaneously into the left and right flanks, respectively. At the time that tumor volume reached 100 mm$^3$, $8 \times 10^5$ CFSE (Invitrogen) labeled OT-1 CD8$^+$ T cells were injected intravenously on day 8, 11 and 14. And on day 17, heart, liver, spleen, lung, kidney, thymus and tumor were excised and analyzed by IVIS imaging system (PerkinElmer).

## cDNA expression, shRNA and CRISPR-Cas9-mediated gene knockout

cDNAs for the studied genes were cloned into pSin-based lentiviral expression vector. shRNA sequences for target genes were inserted into pLKO.1 plasmid. sgRNA oligonucleotides were cloned into the lentiCRISPR vector. The sequences for shRNAs and sgRNAs were listed in Supplementary Data 1. For preparation of lentivirus, pSin-based expression constructs, pLKO.1-based RNAi plasmids or lentiCRISPR-mediated gene editing vectors were co-transfected into HEK293T cells with the psPAX2 and pMD2.G plasmids at a ratio of 2:2:1 by Lipofectamine 2000 (Thermo Fisher Scientific, 11668027) following the manufacturer's instructions. The virus particles were harvested at 48 and 96 h after transfection, filtered by 0.45 µM filter unit (Millipore), and then stored at −80 °C in aliquots.

Cells were infected with lentivirus in the presence of polybrene (4 µg/ml, Sigma-Aldrich, H9268) for 24 h and selected with puromycin (BBI, A610593) for 2 days to eliminate nontransduced cells before subsequent analysis. For gene knocking out, after 2 days of puromycin selection, cells were transferred into a new dish and cultured without puromycin at low density (1000 or 2000 per dish) so that single cell could grow into colonies. After a few days, the single colonies were picked individually and transferred to a new dish. The knockout efficiency was identified by genotyping and Sanger sequencing and further validated for protein absence by immunoblotting.

## Ex vivo Ova-specific cytotoxic CD8$^+$ T-cell killing assay

CD8$^+$ T cells were isolated from spleen of OT-1 mice using the EasySeq mouse CD8$^+$ T-cell isolation kit (STEMCELL, 19858) according to the manufacturer's protocol. And then, CD8$^+$ T cells were co-cultured with Flag-Ova-B16F10 cells at the E:T ratio of 10:1 in 96-well plates. After 12 or 24 h, culture supernatants were collected for LDH release assay (Abcam, ab65393), and cells were collected for flow cytometry analyses.

## CARE-LASS (calcein-release-assay)

$1.5 \times 10^5$ Flag-Ova-B16F10 cells were incubated with 25 µM Calcein AM (Thermo, C1430) at 37 °C for 30 min, and then washed twice with PBS. These labeled Flag-Ova-B16F10 cells were co-cultured with OT-I CD8$^+$ T cells at 37 °C for 6 h. After that, culture supernatants were collected and transferred into new wells to record fluorescence (F value in the formula) using a 485 nm excitation filter and a 515 nm emission filter. The percentage of calcein release was calculated using the following formula: Calcein release (%) = $100 \times (F_{CTL\ assay} - F_{spontaneous\ release})/(F_{total\ lysis} - F_{spontaneous\ release})$.

## RNA extraction, reverse transcription and qPCR

Total RNAs were extracted by TRIzol (Invitrogen) following the manufacturer's instructions and quantified by Nonodrop (Thermo fisher). RNA was reverse-transcribed by using Hiscript III Reverse Transcriptase (Vazyme, R302-01). Quantitative PCR (qPCR) assays were performed in BIO-RAD CFX96 via utilizing AceQ qPCR SYBR Green Master Mix (Vazyme, Q511-02). The detailed primer sequences are exhibited in Supplementary Data 1.

## Flow cytometry analysis

Tumors were excised and mechanically minced, and treated with collagenase (0.2 mg/ml, Sigma) and DNase I (1U/ml) for 15 min at 37 °C. Cells were passed through 70 µm filter. And then leukocytes were separated by performing percoll density gradient centrifugation, and red blood cells were lysed with ACK lysis buffer (0.15 M $NH_4Cl$, 10 mM $KHCO_3$, 0.1 mM $Na_2 \cdot EDTA$, pH7.2). After that, leukocytes were incubated with indicated fluorescent antibodies in the dark for 30 minutes at 4 °C.

For intracellular staining, leukocytes were fixed with 4% paraformaldehyde for 20 min, and then permeabilized with FACS buffer (1% BSA and 0.05% saponin in PBS) for 30 min at 4 °C in the dark. After that, leukocytes were incubated with indicated antibodies in washing buffer (3% FBS and 0.025% saponin in PBS) for 30 min at 4 °C and avoid light. An example of gating stratigies for in vitro and in vivo phenotypic and functional analyses of CD8$^+$ T cells is shown in Supplementary Fig. 8. Antibodies used in flow cytometry are provided in Supplementary Data 1. The flow cytometry data was analyzed by FlowJo software.

## Locus-specific DNA methylation analysis

The locations of CpG islands that could be methylated in the promoter region of indicated genes were predicted according to the information provided by Broad Institute Cancer Cell Line Encyclopedia. And then, according to the prediction, bisulfite specific primers were designed and the sequences were provided in Supplementary Data 1. Genomic

DNA was extracted, and then converted using EpiTect Bisulfite Kit (Qiagen, 59104) following the manufacturer's protocol. The specific bisulfite-converted region was amplified using nest PCR. The resultant PCR products were purified and cloned into pGEM-T Easy vector, and then transformed into DH5α. The positive clones were selected by Sanger sequencing.

## ChIP
Chromatin immunoprecipitation (ChIP) were carried out as previously described[85]. Briefly, cells were fixed with 1% formaldehyde for 10 min at room temperature and then quenched by 0.125 M glycine. The cross-linked lysate was sonicated to fragment DNA to a size of 500–700 bp with a Misonix 3000 ultrasonic cell disruptor. The sonicated chromatin samples were used for immunoprecipitation with indicated antibodies.

## Protein extraction and immunoblotting
For total proteins extraction, cells were washed twice with cold PBS and lysed with lysis buffer containing protease inhibitor on ice. For nuclear proteins extraction, cells were lysed with Dignam buffer A (10 mM Tris·HCl, pH7.4, 10 mM KCl, 1.5 mM MgCl$_2$) on ice for 15 minutes. And then, lysates were centrifuged at $500 \times g$ for 5 min at 4 °C. The crude nuclei pellets were lysed with lysis buffer containing protease inhibitor on ice. The extracted proteins were analyzed by western blot assay with indicated antibodies.

## Immunofluorescence
Cells were washed with PBS, fixed in 4% paraformaldehyde for 15 min at room temperature and then permeabilized with PBS containing 0.25% Triton X-100 for 10 min at room temperature. After that, cells were blocked with 1% BSA in PBS for 1 hour to block the nonspecific binding of the antibody. The negative control cells were treated with RNase III (NEB, cat#E6146) according to the manufacturer's instructions. Cells were incubated with monoclonal anti-dsRNA antibody J2 (Scicons, 10010200) at 4 °C overnight. After incubation, cells were washed with PBST three times and then incubated with secondary antibody (Alexa Fluor 488 goat anti-mouse IgG, Invitrogen, #A11001) for 1 h at room temperature in the dark. Incubate cells with 250 ng/ml DAPI for 10 min and mount coverslip.

## RNA-seq and ChIP-seq Library preparation
GFP$^+$ sgVector, *Mll3*$^{-/-}$ or *Mll4*$^{-/-}$ B16F10 tumor cells ($5 \times 10^5$) were implanted subcutaneously into the right flank of 6- to 8-week-old C57BL/6J male mice. On day 14 post inoculation, tumors were excised and mechanically minced, and treated with collagenase (0.2 mg/ml, Sigma, USA) and DNase I (1 U/ml, Sigma, USA) for 15 min at 37 °C. Single cells from five sgVector, *Mll3*$^{-/-}$ or *Mll4*$^{-/-}$ tumors were pooled separately and passed through a 70 μm filter for two independent flow cytometric sorting of GFP$^+$ tumor cells followed by RNA extraction. rRNA-depleted RNA-seq libraries were prepared using VAHTS Total RNA-seq (H/M/R) Library Prep Kit for Illumina (Vazyme, NR603-01) according to the manufacturer's instructions. Library concentrations were determined by Qubit 2.0 Fluorometer with Qubit dsDNA HS Assay Kit (Thermo, Q32851). An equal amount of each library was pooled together for high-throughput sequencing on NovaSeq 6000 (Illumina) platform to generate paired-end reads.

For preparation of ChIP-seq libraries, immunoprecipitated DNA were quantified by Qubit 2.0 Fluorometer with Qubit dsDNA HS Assay Kit ((Thermo, Q32851) and 1 ng DNA was used to generated bar-coded libraries using VAHTS Universal DNA Library Prep Kit for Illumina V3 (Vazyme, ND607-01) following the manufacturer's guidelines. Library concentrations were determined by Qubit 2.0 Fluorometer with Qubit dsDNA HS Assay Kit (Thermo, Q32851). Equal amount of each library was pooled together for high-throughput sequencing on NovaSeq 6000 (Illumina) platform to generate paired-end reads.

## CRISPR/Cas9 screening data analysis
In vitro and in vivo CRISPR/Cas9 screening data were obtained from previously published studies[23,43], and the enrichment or depletion of sgRNAs were plotted using ggplot2 R package (3.3.2).

## RNA-seq analysis
The quality, adapter content and duplication rate of raw paired-end reads were confirmed by FastQC (0.11.9) with default parameters, and reads were trimmed by Cutadapt. Clean reads were aligned to mouse genome mm10 using HISAT (2.2.1), and the quantification of gene expression was calculated by htseq-count (0.11.1).

Differentially expressed gene analyses were performed using edgeR package (3.36.0). Absolute value of log2 fold change more than 1 and FDR less than 0.05 was used as cutoff to determine significantly differentially expressed genes. Gene set enrichment analysis (GSEA) was performed with GSEA (4.1.0), and the gene sets were obtained from MSigDB database.

Raw paired-end reads were aligned to mouse genome mm10 using STAR (2.7.0a) with the parameters "−outFilterMultimapNmax: 500;−outFilterMatchNmin: 35" to obtain multimapping alignments. The analyzeRepeats.pl function from Homer (http://homer.ucsd.edu/homer/ngs/analyzeRNA.html) software was used to calculate read counts for ERVs transcripts from RNA-seq data. Differential expression for ERVs transcripts was performed with edgeR package (3.36.0), and cutoff of "FDR < 0.05" was used to determine significantly differentially expressed ERVs transcripts. Volcano plots of differentially expressed ERVs transcripts were plotted using ggplot2 R package (3.3.2).

## ChIP-seq Analysis
Raw paired-end reads of H3K4me1 and H3K27ac were trimmed by Trim Galore (0.6.4), and then aligned to mouse genome mm10 using Bowtie2 (2.4.3) with default parameters. Samtools (1.7) was used to remove all unmapped reads and keep one assigned position with the best matched score of non-uniquely mapped reads. Next, PCR duplicates were removed using Picard (2.26.5). For subsequent analyses, reads were extended to 200 bp and normalized to Reads Per Kilobase per Million mapped reads (RPKM) to generate bigwig files by deepTools (2.0). The bigwig track files were visualized with Integrative Genomics Viewer (2.11.1). MACS2 (2.2.7.1) was used to call peaks at FDR cutoff of 0.05. ROSE_main.py function from ROSE (http://younglab.wi.mit.edu/super_enhancer_code.html) was used to identify typical- or super-enhancers based on H3K27ac ChIP-seq signals with parameters of "-s 4000 -t 2500". ROSE_geneMapper.py function was used to annotate enhancers by their nearest genes. The expression difference of these genes between indicated groups was identified by a two-sided Mann–Whitney U test.

Heatmaps and average intensity curves of ChIP-seq signals for H3K27ac and H3K4me1 on typical- or super- enhancer regions were generated by deepTools (2.0).

## TCGA and CCLE data analysis
Gene expression profiles from TCGA and CCLE (Cancer Cell Line Encyclopedia) databases were used for correlation analysis. The Spearman's correlation coefficients and p value of indicated gene pairs were calculated by the R function "cor.test" with the Spearman method.

The patient overall survival and gene expression profiles from the TCGA database were used for survival analysis. Gene expression profiles were normalized by Z-score transformation so that each gene had a mean of 0 and a standard deviation (SD) of 1.

All patients in the TCGA SKCM dataset were divided according to the expression of CD8 [(CD8A + CD8B)/2] into high infiltration group

(>1 S.D.) and low infiltration group (<1 S.D.). For the survival analysis, each group of patients was further split into two subsets according to the optimal expression value calculated by the maxstat R package)[43]. P values were calculated using a log-rank test.

For comparison of immune filtration and function between human tumors with and without alterations in *MLL3* or *MLL4*, genetic information for both genes were extracted from TCGA (MC3 MAF v0.2.8 file), which is generated by the PanCancer Atlas consortium. The mutations were further filtered to eliminate artifacts and reduce false positive rates as previously described[86]. Briefly, "FILTER" needs to be native_wga_mix, WGA or PASS, and only non-silent mutations were retained ("Variant_Classification" should be Frame_Shift_Del, Frame_Shift_Ins, In_Frame_Del, In_Frame_Ins, Missense_Mutation, Nonsense_Mutation, Nonstop_Mutation, Splice_Site, or Translation_Start_Site). Mutations need to be called by at least two software ("NCALLERS" > 1). All patients in the TCGA datasets were divided into two groups based on the *MLL3* or *MLL4* mutation status, and the expression levels of indicated genes between the two groups were compared. *P* values were calculated by the Wilcoxon test.

### Signature-score and association analysis

To investigate the correlation between pyroptosis pathway and immunotherapy response and survival in cancer patients, we calculated a signature score of the pyroptosis pathway for each sample, following previously published method[67,87]. Specifically, Z-score of log2-transformed gene expression for each patient was calculated. Patients were divided into the respondent group (R, manifesting a complete response, partial response, stable disease, or mixed response) and the non-respondent group (NR, showing progressive disease) based on their immunotherapy responses. Then the differential expression between these two groups was identified using a two-sided *t* test. The weighted and normalized sum of the Z-scores for each patient was used as the pyroptosis S-score. Welch two-sample *t* test was used to test the difference in S-score between the R group and NR group.

To test the association of the pyroptosis S-score with survival, the median of S-score was used as a cutoff to divide patients into two groups. R package "survival" was used for survival analysis, and the difference between these two groups was identified by log-rank test.

### Statistics and reproducibility

All experiments were repleated at least twice with similar results. Graphpad Prism 6 was used to make histograms, tumor growth, mice survival curves and to determine statistical significane of each comparison.

### Reporting summary

Further information on research design is available in the Nature Research Reporting Summary linked to this article.

## Data availability

The raw and processed ChIP-seq and RNA-seq datasets generated in this study have been deposited into Gene Expression Omnibus (GEO) under the series entry number of GSE192714 and can be downloaded from the link below: https://www.ncbi.nlm.nih.gov/geo/query/acc.cgi?acc=GSE192714. The in vitro and in vivo CRISPR/Cas9 screening data reanalyzed in this study are available as Supplementary Materials in Pan et al. (https://www.science.org/doi/suppl/10.1126/science.aao1710/suppl_file/aao1710_pan_sm.pdf)[43] and Griffin et al. (https://static-content.springer.com/esm/art%3A10.1038%2Fs41586-021-03520-4/MediaObjects/41586_2021_3520_MOESM3_ESM.xlsx)[23]. The genomic sequencing datasets used for analyzing the association of *MLL4* mutational status with clinical response to PD-L1-PD1 blockade therapies were available as supplementary materials in Hugo et al. (https://www.science.org/doi/suppl/10.1126/science.aad0095/suppl_

file/tables1.mutation_list_all_patients.xlsx, https://www.science.org/doi/suppl/10.1126/science.aad0095/suppl_file/tables2.clinical_and_genome_characteristics_each_patient.xlsx)[44], Mariathasan et al. (http://research-pub.gene.com/IMvigor210CoreBiologies/packageVersions/IMvigor210CoreBiologies_1.0.0.tar.gz)[45], and Snyder, et al. (https://github.com/cetienn01/Multi-Omic-aPDL1)[46]. Data used for analyzing the association of pyroptosis-related gene expression and pyroptosis S-score with therapeutic efficacy of anti-CTLA-4 and anti-PD-1 treatment in cancer patients were available as Supplementary Materials in Snyder et al. (https://cbioportal-datahub.s3.amazonaws.com/skcm_mskcc_2014.tar.gz)[66], Liu et al. (https://github.com/vanallenlab/schadendorf-pd1)[68], Gide et al. (http://tide.dfci.harvard.edu/download/release/Gide2019_PD1_Melanoma_RNASeq.tar.gz/)[69], Kim et al. (http://tide.dfci.harvard.edu/download/release/Kim2018_PD1_Gastric_RNASeq.tar.gz)[70], and Riaz et al. (https://github.com/riazn/bms038_analysis)[71]. TCGA and CCLE datasets were downloaded from https://portal.gdc.cancer.gov/ and https://sites.broadinstitute.org/ccle/datasets, respectively. Oligonucleotides sequence, antibody source and dilution, software, and additional reagents are listed in Supplementary Data 1. All remaining data associated with this study are available within the Article, Supplementary Information. Source data are provided with this paper.

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

## Acknowledgements

We are grateful to G. Li at the Institute of Biophysics, Chinese Academy of Sciences for his critical reading and helpful discussion of the manuscript. We also appreciate H. Tang at Tsinghua University for providing B16F10 and MC38 cell lines. These studies were supported by starting funds of Tianjin Medical University, the National Natural Science Foundation of China (31872825, 32022013, 32270650), and Tianjin Natural Science Foundation (18JCYBJC42400, 20JCJQJC00290) to D.H., the Youth Program of National Natural Science Foundation of China (32000469 to X.G and 32100466 to S.H) and the China Postdoctoral Science Foundation (2021M692408 to S.H).

## Author contributions

D.H. and X.G conceived the project and supervised the study. H.N. performed the majority of the animal and in vitro cell line-based experimental work. S.H. performed ChIP-seq experiment and H.Y. performed flow cytometry analyses. J.J. made libraries for ChIP-seq and RNA-seq. Y.L., R.Z., Z.H., J.L., and K.G. performed bioinformatics analyses under the supervision of X.G. J.Y, Z.L., and Y.B. assisted data interpretation. D.H., S.H., and H.N. drafted the paper with the input from all authors. All authors contributed to editing the paper.

## Competing interests

The author declares no competing interests.

## Additional information

[1]The Province and Ministry Co-sponsored Collaborative Innovation Center for Medical Epigenetics, State Key Laboratory of Experimental Hematology, Key
Laboratory of Immune Microenvironment and Disease of Ministry of Education, Department of Cell Biology, School of Basic Medicine, Tianjin Medical
University, Tianjin, China. [2]State Key Laboratory of Experimental Hematology, National Clinical Research Center for Blood Diseases, Haihe Laboratory of Cell
Ecosystem, Institute of Hematology & Blood Diseases Hospital, Chinese Academy of Medical Sciences & Peking Union Medical College, Tianjin, China. [3]Key
Laboratory of Immune Microenvironment and Disease of Ministry of Education, Department of Biochemistry and Molecular Biology, School of Basic Medical
Sciences, Tianjin Medical University, Tianjin, China. [4]Key Laboratory of Immune Microenvironment and Disease of Ministry of Education, Department of
Immunology, School of Basic Medical Sciences, Tianjin Medical University, Tianjin, China. [5]Key Laboratory of Cancer Prevention and Therapy, Tianjin's Clinical
Research Center for Cancer, National Clinical Research Center for Cancer, Tianjin Medical University Cancer Institute and Hospital, Tianjin, China. [6]These
authors contributed equally: Hanhan Ning, Shan Huang, Yang Lei. ✉e-mail: gaoxin1@ihcams.ac.cn; hudq@tmu.edu.cn

