## [Peer Review File · Nature Communications]

Enhancer decommissioning by Mll4 ablation elicits dsRNA-interferon signaling and Gsdmd-mediated pyroptosis to potentiate anti-tumor immunityReviewers' Comments:

Reviewer #1:

Remarks to the Author:

Enhancer decommissioning by MLL4 ablation elicits dsRNA-interferon signaling and Gsdmd-mediated pyroptosis to potentiate anti-tumor immunity

Suggestion: Accept with minor corrections listed below

A. Summary:

Here, the authors demonstrate that deletion of MLL4 or MLL3 in tumor cells promotes anti-tumor immunity. This effect has been observed before in at least one other model (detailed more below) but is an idea expanded on here with additional model systems and in great mechanistic detail. In particular, the authors go to great experimental lengths to demonstrate that mechanistically MLL4 loss mediates immune modulatory effects through multiple pathways. This includes activation of double-stranded RNA stress that activates interferon responses as well as activation of cell death, most notably pyroptosis. The connection to pyroptosis is particularly novel. The authors go on to nicely characterize how these pathways are activated as related to the enhancer regulatory functions of MLL4. Specifically, they find that MLL4 deletion results in enhancer decommissioning and subsequent lost expression of Ago2 (a component of the RISC complex) and DNA methyltransferases. They go to functionally validate that these effects showing they are a reasonable explanation for their transcriptional data and the activation of pyroptosis.

Overall, the authors have put together an excellent manuscript that will be of significant interest to those in the field of cancer epigenetics. The impact of this work is high, particularly given the implications of stratifying patients with MLL4 mutations as good candidates for ICB therapies, although this was an idea first put forth by others. Its impact is further limited by a heavy reliance on B16 cells for a majority of the work. Nonetheless, the findings presented here further substantiate and significantly build on previous ideas put forth while offering novel mechanistic insight. The mechanistic work is impressive and comprehensive, employing multiple techniques, knockouts, and modes of evidence. The methods are detailed and lends themselves to reproducibility.

B. More detailed comments:

- Experiments are very thorough and well controlled. No additional experiments are suggested to support the current work or conclusions. However, the authors could do a better job of emphasizing the limitations of the work given that some of these mechanisms may have a context-dependency, and this study relied on a single melanoma cell line for the vast majority of the experiments. Furthermore, MLL4/KMT2D mutations more commonly occur in other epithelial cancers like bladder cancers and squamous cell carcinomas, which are very different in many ways from melanomas biologically, so unless shown, the underlying mechanisms may be different.
- Previous literature using CRISPR-GEMM models identified MLL4 (KMT2D) loss of function mutations as a modulator of immune responses in cancer and during immune checkpoint blockade (ICB) (Wang et al. Cancer Discovery 2020 Dec 10(12): 1912-1933; PMID: 32887696). Authors Ning & Huang et al add evidence to support this work in their manuscript while making additional interesting and compelling mechanistic connections to pyroptosis. The overall novelty of Ning & Huang et al's work is bit undercut by the previous findings of Wang et al, but remains important and adds new, thought-provoking elements to the roles of MLL4 in immune modulation in cancer and will be of interest to the readers of Nature Communications. However, we do ask the authors to add references such as Wang et al. to their work to put it better into context of existing literature and to amend certain wording throughout the paper overstating its novelty. For example:
 - o Lines 104-107 claiming "whether... MLL3 and MLL4-regulated enhancers are involved in tumor

immune response and therapeutic resistance to checkpoint blockades is still unexplored" should be altered.

- Methodology is appropriate and utilizes the relevant cellular and mouse models.
- The methods are given in excellent detail and at the level needed for reproducibility. The only exception to this is the absence of the cited Table S1 said to contain primer sequences and antibodies. This table should be added to the final version of the manuscript.
- It will be helpful if the authors, in the introduction or discussion, briefly discuss their thoughts on the pervasive nature of MLL4/3 mutations in human cancer and why there is such a selective pressure for these mutations in light of the authors data showing that these mutations actually help to clear tumors via immunomodulation. A clear idea or hypothesis should be put forth concerning this point as well as a more detailed discussion of the clear tumor suppressive functions of MLL4 in cancer. For example, several studies have shown that MLL4 mutations can actually promote early clonal expansion in epithelial tumors like bladder cancer:

Li R, Du Y, Chen Z, Xu D, Lin T, Jin S, Wang G, Liu Z, Lu M, Chen X, Xu T, Bai F. Macroscopic somatic clonal expansion in morphologically normal human urothelium. *Science*. 2020 Oct 2;370(6512):82-89. doi: 10.1126/science.aba7300. PMID: 33004515.

And skin:

Fowler JC, King C, Bryant C, Hall MWJ, Sood R, Ong SH, Earp E, Fernandez-Antoran D, Koeppl J, Dentre SC, Shorthouse D, Durrani A, Fife K, Rytina E, Milne D, Roshan A, Mahububani K, Saeb-Parsy K, Hall BA, Gerstung M, Jones PH. Selection of Oncogenic Mutant Clones in Normal Human Skin Varies with Body Site. *Cancer Discov*. 2021 Feb;11(2):340-361. doi: 10.1158/2159-8290.CD-20-1092. Epub 2020 Oct 21. PMID: 33087317; PMCID: PMC7116717.

Furthermore, rather than suppressing pyroptosis as presented here, another recent study showed MLL4 was important for the promotion of ferroptosis, another form of regulated cell death:

Egolf S, Zou J, Anderson A, Simpson CL, Aubert Y, Prouty S, Ge K, Seykora JT, Capell BC. MLL4 mediates differentiation and tumor suppression through ferroptosis. *Sci Adv*. 2021 Dec 10;7(50):eabj9141. doi: 10.1126/sciadv.abj9141. Epub 2021 Dec 10. PMID: 34890228; PMCID: PMC8664260.

These references should be cited and the discrepancies should be at least discussed and mentioned in the Discussion.

Reviewer #2:

Remarks to the Author:

In this manuscript Ning, Huang, Lei et al evaluate the impact of MLL3 and MLL4 genetic ablation on tumor immunogenicity in mouse syngeneic tumor models. First they show that MLL3 or MLL4 knockout enhances OT-1 T cell killing of Ova expressing target cells, and then confirm enhanced immunogenicity in the B16 model in vivo, which they prove is dependent on enhanced T cell immunity. Using human TCGA data, they show that low MLL3/4 expression correlates with increased GZM/CD8 gene expression in several tumor types, and predicts improved outcome in CD8 high tumors. They then explore the mechanistic basis for these findings, discovering increased IFN gene and MHC1 gene expression, coupled with de-repression of transposal elements and dsRNA. They also show that MLL4 correlates with Ago2 expression in the TCGA, that MLL4 KO decreases Ago2 expression

in B16 cells, and that that Ago2 over-expression in this setting can partially impair the enhanced immunogenicity. They also noted particularly strong upregulation of Gsdmd in Mll4 KO B16 cells, which they also show enhances pyroptosis and immunogenicity in vivo. They then link Dnmt1 and Dnm3a to Gsdmd and Casp1/4 enhancer control, showing that Mll4 KO decreases Dnmt1/3a expression which promotes their de-repression. Finally, they demonstrate that immunogenicity is further enhanced in vivo with concomitant PD-1 blockade.

Overall, this is a comprehensive manuscript that convincingly demonstrates a role for MLL3 and MLL4 in restraining tumor immunogenicity in mice. My primary critique is that, other than the associations by TCGA/CCLE there is no evidence that the same pathways are relevant in human cancer cells. They should at least repeat a subset of studies in human melanoma cancer cell lines (or other MLL4 high cell lines identified in their CCLE analyses) and demonstrate that the core conclusions of the manuscript are relevant in human cancer cells. Namely that MLL4 KO decreases human AGO2 and DNMT1/3A expression, with evidence for de-repression of dsRNA and GSDMD. Also does this boost an IFN gene program, MHC I and PD-L1 expression in human cancer cells?

Reviewer #3:

Remarks to the Author:

Epigenetic regulators represent an emerging layer of players in controlling tumor response to host immunity and immunotherapy. This study by Ning and colleagues identifies MLL3/4 as new epigenetic players in tumor immunity. By interrogating published CRISPR screen results, the authors discovered MLL3/4 as promising candidates for further functional validation and mechanistic investigation. Using in vitro coculture assay and in vivo animal models, the authors demonstrated a suppressive role of tumor cell-intrinsic MLL3/4 in CD8+ T immunity, supported by evidence that either MLL3 or MLL4 depletion in B16 tumor model led to an enhanced T cell cytotoxicity and an increased intratumoral T cell infiltration. They further explored the molecular mechanisms and identified two important pathways, i.e., endogenous dsRNA-induced IFN-I response and GSDMD-mediated pyroptosis, both known to inflame tumors. Further, they showed that enhancers associated with AGO2 and DNMTs are direct targets of MLL3/4, and upon MLL3/4 loss, decreased AGO2 and DNMTs led to elevated dsRNAs and de-repressed GSDMD expression, respectively. On those bases, the authors tested the combination of MLL4 KO with PD1 blockade therapy and saw a cooperative effect on B16 tumor growth control.

Overall, this is a comprehensive and compelling study. A few pieces of interesting findings distinguish this study from previous reports, including the concurrent activation of two immunogenic pathways by perturbing a single enhancer regulator and the new identification of GSDMD in the tumor response to T cell cytotoxicity. A few specific comments are list below.

- 1) Since MLL3 and MLL4 are frequently mutated in human cancers, leading to loss of function, do those tumors with MLL3/4 mutation have fewer T cell infiltration and poorer response to immunotherapy?
- 2) NLRP1 is recently reported to sense dsRNA and activate GSDMD. Is there any connection between dsRNA-induced IFN-I response and GSDMD-mediated pyroptosis? The authors may at least discuss this.
- 3) I feel the sentence related to Figure S1A is ambiguous. Figure S1A can be described more clearly.
- 4) In Figures 1B and 1C, do Mll3 KO and Mll4 KO B16 cells grow normally? What is the basal level of LDH release in those cells compared with WT cells in the absence of OT1 cells?
- 5) Flow cytometry data related to GzmB and IFN γ staining should be improved, as they appear not in a typical pattern.
- 6) A high E: T ratio of 10:1 was used in the killing assay, likely because OT1 cells are not activated and thus not very effective before being applied to tumor cell layers. In tumor masses, tumor cells usually numerically overwhelm T cells.

- 7) When talking about "parental", did the author refer to cells without any genetic modifications or cells also modified by CRISPR/Cas9 with control sgRNA? The use of "Parental" should be avoided if the latter is the case. Capital letters should be used when talking about proteins, for example MLL3, DNMT1 and so on.
- 8) In heatmaps of Figure 3, n=5 was claimed in the legends, but only two replicates can be clearly spotted from the figures.
- 9) Will the concurrent depletion of dsRNA and pyroptosis pathways completely rescue the growth defect of Mll3 and Mll4 KO tumors in immunocompetent mice?
- 10) The cartoon in the last Figure should be modified, because naïve T cells are mainly primed in the lymphoid tissues and recruited to the tumor sites, rather than be primed locally in tumor masses, although sometimes local priming occurs for example in TLS.

Reviewer #4:

Remarks to the Author:

In this manuscript, Ning, Huang and Lei et al. investigated how depletion of Mll3 and Mll4, two enhancer-associated H3K4 mono-methyltransferases, affects the tumor microenvironment and anti-tumor immunity. They present large amounts of data to claim that both dsRNA-interferon pathway and pyroptotic cell death are important for increased anti-tumor immune responses observed in Mll4 deficient tumors. However, some critical data regarding pyroptosis are not convincing, and some are missing to support their claims. Here are my concerns about this manuscript:

Major concerns:

- 1) The author co-incubated CD8 OT-1 cells and B16-Ova in vitro to detect pyroptosis in B16-Ova cells with various genetic manipulations (Fig.4 and 5). These images should be the most convincing data because "seeing is believing." However, the data presented is disappointed. Pyroptotic cells usually develop giant membrane balloons, but I basically can't see any cells exhibiting pyroptotic morphology in Gsdmd-WT cells (Fig. 4, in four indicated cells, only one cell shows pyroptotic morphology but random necrosis can't be ruled out in this cell) and in Mll4KO shGFP cells (Fig.5). The Mll4KO shGFP cells actually exhibit the morphology of apoptosis (shrunk cells with small blebbing). What makes it worse is that there is no cell death in any other B16 cells with low GSDMD expression in Fig.4 or Fig.5. But we know it is not the case. CD8 T OT-1 cells will kill B16-Ova cells regardless of GSDMD expression. The GSDMD may switch the types of cell death (e.g., from apoptosis to pyroptosis), but cell death should be observed in B16-Ova cells after the CD8 cell challenge.
- 2) In the immunoblots about GSDMD (Fig. S4C, S4D, S6J), there are GSDMD-NT fragments detected in all samples. It is surprising because GSDMD-NT generation by protease cleavage is an indicator of pyroptosis, and the data suggest that pyroptosis occurs in all the cells. There must be something wrong with the data or the interpretation of the experiments (the GSDMD-NT band could be an unspecific band).
- 3) The mechanism underlying this CD8-triggered GSDMD pyroptosis is unclear and the claimed pyroptosis is not consistent with previous studies. The authors claim that GSDMD is activated by CD8 T cells, and the cleavage site is D276 (cleavage site of Caspase-1/4/5/11). CD8 T cells induce tumor cell death mainly by delivering granzymes. Previous studies indicate that GSDMB and GSDME are cleaved and activated by GzmA and GzmB, respectively. But the Gzm pathways may not fit the current case because GzmB is the only known granzyme that cuts after aspartic acid, but it has been proved that GSDMD is not a substrate of GzmB or any other Gzms (Ref.40 Fig S2C). Moreover, NK cells don't induce pyroptosis in cells expressing GSDMD (Ref.40), which is inconsistent with current data. To support their claims, the authors need to:

- a) Demonstrate that inflammasome-activated pyroptosis could occur in B16. Inflammasome-activated pyroptosis mainly occurs in macrophages and some epithelial cells, not in other cells, because inflammasome components are selectively expressed. I can't find any paper showing that B16 cells undergo inflammasome-activated pyroptosis. Therefore, the authors need to prove it. They may try some classical pyroptosis treatments (e.g., LPS+ATP and Poly dA:dT for canonical inflammasome; LPS electroporation for non-canonical inflammasome) to show that inflammasome assembly and pyroptosis occur in B16.
- b) Show caspase-1 or caspase-11 activation (cleavage), PI uptake, pyroptotic morphology and other indicators of pyroptosis in B16 cells treated with CD8 T cells.
- c) Find out how CD8 T cells induce B16 pyroptosis.

Minor concerns:

- 1) As far as I know, the counterpart of human caspase-4 is mouse caspase-11, and there is no mouse caspase-4, or researchers usually don't use this name.
- 2) The authors need to distinguish between human and mouse proteins. The statement in line 110-111 "The Gasdermin (Gsdm) family of pore-forming proteins that comprise Gsdma, Gsdmb, Gsdmc, Gsdmd, and Gsdme" is wrong. Human and mouse proteins share the same names, such as GSDMD; you can tell them using hGSDMD and mGSDMD. If referred to genes, use human GSDMD and mouse *Gsdmd* (italicized). More importantly, the human GSDM family contains GSDMA, GSDMB, GSDMC, GSDMD, GSDME and DFNB59. But mouse GSDM family includes Gsdma 1-3, Gsdmc 1-4, Gsdmd, Gsdme and Dfnb59. The authors studied the mouse GSDM family throughout the manuscript, but they never mentioned this information or never stated it correctly. In figure 4A and S4A, the authors must change the name Gsdma to Gsdma1, and Gsdmc to Gsdmc1.
- 3) Figure legend 1B, ET ratio 1:10 should be 10:1

Responses to the Reviewers' Comments

Thanks for all the reviewers' comments on our manuscript. We are glad to read the constructive suggestions aimed at improving the quality of this work. Based on their insightful advice, we have revised the manuscript in terms of additional experimental studies, modifications of text and figures, and added extra discussions on our findings. Below are the detailed point-by-point responses to the reviewers' queries.

REVIEWER COMMENTS

Reviewer #1, expertise in epigenetics and cancer (Remarks to the Author):

Enhancer decommissioning by Mll4 ablation elicits dsRNA-interferon signaling and Gsdmd-mediated pyroptosis to potentiate anti-tumor immunity

Suggestion: Accept with minor corrections listed below

A. Summary:

Here, the authors demonstrate that deletion of MLL4 or MLL3 in tumor cells promotes anti-tumor immunity. This effect has been observed before in at least one other model (detailed more below) but is an idea expanded on here with additional model systems and in great mechanistic detail. In particular, the authors go to great experimental lengths to demonstrate that mechanistically MLL4 loss mediates immune modulatory effects through multiple pathways. This includes activation of double-stranded RNA stress that activates interferon responses as well as activation of cell death, most notably pyroptosis. The connection to pyroptosis is particularly novel. The authors go on to nicely characterize how these pathways are activated as related to the enhancer regulatory functions of MLL4. Specifically, they find that MLL4 deletion results in enhancer decommissioning and subsequent lost expression of Ago2 (a component of the RISC complex) and DNA methyltransferases. They go to functionally validate that these effects showing they are a reasonable explanation for their transcriptional data and the activation of pyroptosis.

Overall, the authors have put together an excellent manuscript that will be of significant interest to those in the field of cancer epigenetics. The impact of this work is high, particularly given the implications of stratifying patients with MLL4 mutations as good candidates for ICB therapies, although this was an idea first put forth by others. Its impact is further limited by a heavy reliance on B16 cells for a majority of the work. Nonetheless, the findings presented here further substantiate and significantly build on previous ideas put forth while offering novel mechanistic insight. The mechanistic work is impressive and comprehensive, employing multiple techniques, knockouts, and modes of evidence. The methods are detailed and lends themselves to reproducibility.

Response: We would like to thank the reviewer for his/her very positive remarks on our study and for pointing out the weaknesses of the current work, from which we can continue to improve the quality of this manuscript and thus the impact of our study. To test whether our mechanistic findings, which were largely revealed through mouse B16 melanoma cells and computational analyses of associations using human CCLE

and TCGA databases, also apply to human tumor cells, we depleted MLL4 in several human cancer cell lines of diverse tissue origin and found the molecular pathways that mediate the effects of MLL4 loss on tumor immune response are essentially conserved between mouse and human tumor cells. These results were incorporated into the updated figures and shown as Fig. 2q, 2r, 2s, supplementary Fig.3d, 3f, 3l, 3m, 3o, 3r and 6d. The legends for these newly inserted panels were added to the revised manuscript text as well.

B. More detailed comments:

• Experiments are very thorough and well controlled. No additional experiments are suggested to support the current work or conclusions. However, the authors could do a better job of emphasizing the limitations of the work given that some of these mechanisms may have a context-dependency, and this study relied on a single melanoma cell line for the vast majority of the experiments. Furthermore, MLL4/KMT2D mutations more commonly occur in other epithelial cancers like bladder cancers and squamous cell carcinomas, which are very different in many ways from melanomas biologically, so unless shown, the underlying mechanisms may be different.

Response: We thank the reviewer for complimentary notes on our experimental design and technique robustness, and fully agree that the findings of our work may have some limitations and are context-dependent. To fulfill our interest and also to provide answers to reviewer 2's comments, we examined the transcript levels of *AGO2*, *PD-L1*, a few selected interferon-responsive genes, and genes encoding MHC-I by RT-qPCR, and conducted immunoblotting analyses of the protein levels of DNMT3A and DNMT1 in two control- and MLL4-depleted human melanoma (A375 and SK-MEL-28) and two colorectal (DLD-1 and H1299) cell lines, respectively. In line with the strong correlations between *MLL4* expression with levels of these factors in human cancers revealed by computational analyses of TCGA/CCLL databases, MLL4 depletion markedly decreases the expression levels of *AGO2*, *DNMT1*, and *DNMT3A* while mRNA levels of the selected interferon-responsive genes and MHC-I machinery are significantly elevated in most of these tested human cancer cell lines, indicating that our mechanistic findings, to large extent, also apply to some of the human cancer cells. However, due to the substantial variations in the genetic and epigenetic background of different cancer types, we can envision that loss of *MLL4* expression may not trigger dsRNA-interferon signaling and GSDMD-mediated pyroptosis in cancers that bear a loss of function mutations in *DNMT3A* as transcriptional reduction of inactive DNMT3A is not functionally meaningful. These new results from human cancer cell lines and their legends have been incorporated into the revised figures and manuscript (Fig. 2q, 2r, 2s, supplementary Fig.3d, 3f, 3l, 3m, 3o, 3r, and 6d.). We also put forward the limitations of our mechanistic findings and emphasize the genetic and epigenetic influence on the regulatory role of MLL4 in tumor immune response in the discussion section of the revised manuscript (from lines 696 to 700).

• *Previous literature using CRISPR-GEMM models identified MLL4 (KMT2D) loss of function mutations as a modulator of immune responses in cancer and during immune checkpoint blockade (ICB) (Wang et al. Cancer Discovery 2020 Dec 10(12): 1912-1933; PMID: 32887696). Authors Ning & Huang et al add evidence to support this work in their manuscript while making additional interesting and compelling mechanistic connections to pyroptosis. The overall novelty of Ning & Huang et al's work is bit undercut by the previous findings of Wang et al, but remains important and adds new, thought-provoking elements to the roles of MLL4 in immune modulation in cancer and will be of interest to the readers of Nature Communications. However, we do ask the authors to add references such as Wang et al. to their work to put it better into context of existing literature and to amend certain wording throughout the paper overstating its novelty. For example:*

o Lines 104-107 claiming “whether... MLL3 and MLL4-regulated enhancers are involved in tumor immune response and therapeutic resistance to checkpoint blockades is still unexplored” should be altered.

Response: We are grateful to the reviewer for this constructive suggestion, and have now cited and discussed the paper for the first identification of the immune modulatory function of MLL4 through CRISPR-GEMM screen in a murine model of liver cancer. We also rephrased the expression in the revised manuscript to ensure not to overstate the novelty of our findings. “whether... MLL3 and MLL4-regulated enhancers are involved in tumor immune response and therapeutic resistance to checkpoint blockades is still unexplored” is now reading as “whether... MLL3 and MLL4-regulated enhancers are involved in tumor immune response and therapeutic resistance to checkpoint blockades remain incompletely understood” (line 106).

• *Methodology is appropriate and utilizes the relevant cellular and mouse models.*

Response: Thank you!

• *The methods are given in excellent detail and at the level needed for reproducibility. The only exception to this is the absence of the cited Table S1 said to contain primer sequences and antibodies. This table should be added to the final version of the manuscript.*

Response: Thanks to the reviewer for pointing out the missing Table S1. We added it to the revised manuscript as supplementary material and referred to it in the Materials and Methods section.

• *It will be helpful if the authors, in the introduction or discussion, briefly discuss their thoughts on the pervasive nature of MLL4/3 mutations in human cancer and why there is such a selective pressure for these mutations in light of the authors data showing that these mutations actually help to clear tumors via immunomodulation. A clear idea or hypothesis should be put forth concerning this point as well as a more detailed discussion of the clear tumor suppressive functions of MLL4 in cancer. For example, several studies have shown that MLL4 mutations can actually promote early clonal*

expansion in epithelial tumors like bladder cancer:

Li R, Du Y, Chen Z, Xu D, Lin T, Jin S, Wang G, Liu Z, Lu M, Chen X, Xu T, Bai F. Macroscopic somatic clonal expansion in morphologically normal human urothelium. *Science*. 2020 Oct 2;370(6512):82-89. doi: 10.1126/science.aba7300. PMID: 33004515.

And skin:

Fowler JC, King C, Bryant C, Hall MWJ, Sood R, Ong SH, Earp E, Fernandez-Antoran D, Koeppl J, Dentro SC, Shorthouse D, Durrani A, Fife K, Rytina E, Milne D, Roshan A, Mahububani K, Saeb-Parsy K, Hall BA, Gerstung M, Jones PH. Selection of Oncogenic Mutant Clones in Normal Human Skin Varies with Body Site. *Cancer Discov*. 2021 Feb;11(2):340-361. doi: 10.1158/2159-8290.CD-20-1092. Epub 2020 Oct 21. PMID: 33087317; PMCID: PMC7116717.

Furthermore, rather than suppressing pyroptosis as presented here, another recent study showed MLL4 was important for the promotion of ferroptosis, another form of regulated cell death:

Egolf S, Zou J, Anderson A, Simpson CL, Aubert Y, Prouty S, Ge K, Seykora JT, Capell BC. MLL4 mediates differentiation and tumor suppression through ferroptosis. *Sci Adv*. 2021 Dec 10;7(50):eabj9141. doi: 10.1126/sciadv.abj9141. Epub 2021 Dec 10. PMID: 34890228; PMCID: PMC8664260.

These references should be cited and the discrepancies should be at least discussed and mentioned in the Discussion.

Response: The reviewer raised an extremely excellent question that we have been thinking about since we first noted the vital role of MLL3 and MLL4 in antagonizing tumor immunity in 2018. It is well known in the areas of cancer epigenetics and cancer biology that loss of function mutations of *MLL4* is one of the most recurrent genetic alterations across multiple tumor types, including the reviewer's mentioned skin and bladder cancer. Furthermore, the tumor-suppressive function of MLL4 has been well established through studies on the genetic disruption of the *Mll4* gene in a mouse model of several cancer types, such as lymphoma, lung and brain cancers¹⁻⁴. However, it is increasingly recognized that tumors develop and progress through competition between their invasive growth properties, and the immune suppressive pressure of the surrounding tumor microenvironment. We speculated that MLL4 may possess essential roles in regulating multiple aspects of normal cellular function *in vivo*. One of its physiological roles is to promote cellular differentiation to suppress clonal expansion and malignant transformation *in vivo*, thus acting as a tumor suppressor. Another function is to curtail cellular immunogenicity to prevent normal cells from immune attack and clearance to maintain tissue homeostasis. By balancing two aspects of cellular function, MLL4 inactivation could thus either promote tumor development in the context of co-occurring genetic alterations of other genes or unleash tumor immunogenicity to potentiate anti-tumor immunity of tumor

microenvironment to restrain tumor formation and progression. This speculation could not only explain why genetic deletion of *MLL4* alone is incompetent of driving tumor formation and development in multiple mouse tissues but can also provide reasonable explanations for rare detection of loss of function mutations on both alleles of the *MLL4* gene in most human tumor samples of various tissue origins, as inactivation of both alleles could lead to a very robust anti-tumor immune response to prevent tumor formation and development. We have discussed the seemingly contradicting roles of MLL4 in tumor regulation in the revised manuscript from lines 602 to 612.

Reviewer #2, expertise in dsDNA and cancer immunology (Remarks to the Author):

In this manuscript Ning, Huang, Lei et al evaluate the impact of MLL3 and MLL4 genetic ablation on tumor immunogenicity in mouse syngeneic tumor models. First they show that Mll3 or Mll4 knockout enhances OT-1 T cell killing of Ova expressing target cells, and then confirm enhanced immunogenicity in the B16 model in vivo, which they prove is dependent on enhanced T cell immunity. Using human TCGA data, they show that low MLL3/4 expression correlates with increased GZM/CD8 gene expression in several tumor types, and predicts improved outcome in CD8 high tumors. They then explore the mechanistic basis for these findings, discovering increased IFN gene and MHC1 gene expression, coupled with de-repression of transposal elements and dsRNA. They also show that MLL4 correlates with Ago2 expression in the TCGA, that Mll4 KO decreases Ago2 expression in B16 cells, and that that Ago2 over-expression in this setting can partially impair the enhanced immunogenicity. They also noted particularly strong upregulation of Gsdmd in Mll4 KO B16 cells, which they also show enhances pyroptosis and immunogenicity in vivo. They then link Dnmt1 and Dnm3a to Gsdmd and Casp1/4 enhancer control, showing that Mll4 KO decreases Dnmt1/3a expression which promotes their de-repression. Finally, they demonstrate that immunogenicity is further enhanced in vivo with concomitant PD-1 blockade.

Overall, this is a comprehensive manuscript that convincingly demonstrates a role for MLL3 and MLL4 in restraining tumor immunogenicity in mice. My primary critique is that, other than the associations by TCGA/CCLE there is no evidence that the same pathways are relevant in human cancer cells. They should at least repeat a subset of studies in human melanoma cancer cell lines (or other MLL4 high cell lines identified in their CCLE analyses) and demonstrate that the core conclusions of the manuscript are relevant in human cancer cells. Namely that MLL4 KO decreases human AGO2 and DNMT1/3A expression, with evidence for de-repression of dsRNA and GSDMD. Also does this boost an IFN gene program, MHC I and PD-L1 expression in human cancer cells?

Response: We appreciate the reviewer for the positive comments on our work and the constructive suggestions. To assess the relevance of our mechanistic findings revealed using mouse melanoma cells, we depleted MLL4 expression in two cell lines

of human melanoma and colorectal cancer, respectively, where strong correlations of *MLL4* expression with mRNA levels of *AGO2*, *DNMT*, and *DNMT3A* are noted using computational analyses of CCLE and TCGA datasets in our study. Similar to the effects of *Mll4* ablation in mouse B16 cells, *MLL4* knockdown also attenuates the expression of *AGO2*, *DNMT1*, and *DNMT3A* in several of these human cancer cell lines. In addition, levels of dsRNA, GSDMD, PD-L1, and many genes related to the interferon response and antigen presentation by MHC1 are also markedly increased in most of the tested human cancer cell lines depleted for *MLL4* expression as compared to the corresponding control human cells. These results were incorporated into the updated figures and shown as supplementary Fig.3d, 3f, 3l, 3m, 3o, 3r, and 6d. The legends for these newly inserted panels were added to the revised manuscript text as well.

Reviewer #3, expertise in epigenetics, ERV and cancer immunology (Remarks to the Author):

Epigenetic regulators represent an emerging layer of players in controlling tumor response to host immunity and immunotherapy. This study by Ning and colleagues identifies MLL3/4 as new epigenetic players in tumor immunity. By interrogating published CRISPR screen results, the authors discovered MLL3/4 as promising candidates for further functional validation and mechanistic investigation. Using in vitro coculture assay and in vivo animal models, the authors demonstrated a suppressive role of tumor cell-intrinsic MLL3/4 in CD8+ T immunity, supported by evidence that either MLL3 or MLL4 depletion in B16 tumor model led to an enhanced T cell cytotoxicity and an increased intratumoral T cell infiltration. They further explored the molecular mechanisms and identified two important pathways, i.e., endogenous dsRNA-induced IFN-I response and GSDMD-mediated pyroptosis, both known to inflame tumors. Further, they showed that enhancers associated with AGO2 and DNMTs are direct targets of MLL3/4, and upon MLL3/4 loss, decreased AGO2 and DNMTs led to elevated dsRNAs and de-repressed GSDMD expression, respectively. On those bases, the authors tested the combination of MLL4 KO with PD1 blockade therapy and saw a cooperative effect on B16 tumor growth control.

Overall, this is a comprehensive and compelling study. A few pieces of interesting findings distinguish this study from previous reports, including the concurrent activation of two immunogenic pathways by perturbing a single enhancer regulator and the new identification of GSDMD in the tumor response to T cell cytotoxicity. A few specific comments are list below.

Response: We are glad to see the positive remarks and insightful suggestions from the reviewer. As suggested, we performed additional computational analyses and revised the manuscript to make it more explicit to readers and researchers in the fields.

1) Since *MLL3* and *MLL4* are frequently mutated in human cancers, leading to loss of function, do those tumors with *MLL3/4* mutation have fewer T cell infiltration and poorer response to immunotherapy?

Response: We thank the reviewer for putting forward this constructive comment. To address this question, we categorized all TCGA tumor patients into two groups based on the mutational status of *MLL3* or *MLL4* and compared the frequency of infiltrated total and CD8+ T cells as well as their cytotoxicity between two groups of human tumor patients. In line with findings in a murine melanoma model, we found tumors mutated for *MLL3* or *MLL4* have a generally increased abundance of total and CD8+T cells and elevated expression of cytotoxic molecules compared to tumors that contain wild-type *MLL3* or *MLL4*. To explore the association of *MLL3* or *MLL4* mutation with the immunotherapeutic response, we analyzed two cohorts of urothelial cancer patients and one cohort of metastatic melanoma patients who are treated with anti-PD1 and have their tumor exome sequenced before immunotherapy. Tumors with mutations in *MLL4* respond better to anti-PD1 treatment than tumors not mutated for the corresponding gene and thus immune checkpoint blockade therapy have better clinical efficacy in patients bearing *MLL4* mutant tumors. These new results were incorporated into updated Figure 2 as Fig. 2q, 2r, and 2s.

2) *NLRP1* is recently reported to sense dsRNA and activate GSDMD. Is there any connection between dsRNA-induced IFN-I response and GSDMD-mediated pyroptosis? The authors may at least discuss this.

Response: The reviewer raised a very excellent point that may help us address the underlying mechanisms for the activation of GSDMD-mediated pyroptosis in tumor cells upon loss of *Mll4* expression soon. To explore this possibility, we analyzed the RNA-seq expression of canonical inflammasome sensors in control and *Mll4*-ablated B16 cells and found both *Nlrp1a* and *Nlrp1b* are transcriptionally silenced in this cell type, suggesting activation of GSDMD-mediated pyroptosis is not mediated by Nlrp1 sensing of dsRNAs in *Mll4* KO B16 cells. Since transcriptional depression of *GSDMD* and dsRNAs accumulation also occur in human cancer types of diverse tissue origins, it is extremely possible that NLRP1 sensing of dsRNA is involved in the induction of GSDMD cleavage and subsequent pyroptosis in cells expressing NLRP1. We are planning experiments to test this idea in a few human cell lines.

Figure R1. Histogram showing the RNA-seq expression of canonical inflammasome

sensors in control and *Mll4*-deleted B16 cells.

3) I feel the sentence related to Figure S1A is ambiguous. Figure S1A can be described more clearly.

Response: Thanks to the reviewer for pointing out the inexplicit description of Fig. S1A. we have modified the sentence and now it reads “Analysis of an additional CRISPR/Cas9 genetic screen for epigenetic regulators of tumor immunity *in vivo* revealed significant depletion of *Mll4* sgRNAs in both B16 melanoma and Lewis lung carcinoma (LLC1) tumors as well as markedly reduced representation of both *Mll3* and *Mll4* targeting sgRNAs in LLC1 tumors in immunocompetent mice as compared to the corresponding tumors in immunocompromised NSG mice, indicating an immunosuppressive function of tumor cell-intrinsic MLL3 and MLL4 *in vivo* as well.” We also amended other sentences to ensure all figures are clearly described across the whole manuscript.

4) In Figures 1B and 1C, do *Mll3* KO and *Mll4* KO B16 cells grow normally? What is the basal level of LDH release in those cells compared with WT cells in the absence of OT1 cells?

Response: *MLL3* and *MLL4* are recurrently mutated in various types of human tumors and these genetic alterations in most cases are either out-of-frame truncating indels or non-sense mutations. Tissue-specific loss of *Mll3* and *Mll4* has been found to have a synergistic effect with other oncogenic factors in driving tumor development and progression in multiple murine models of human cancers^{1,2,4}. Interestingly, *Mll4* deletion was shown to have a negative effect on human colorectal and medulloblastoma cancer cell proliferation during long-term *in vitro* culture⁵. Likewise, we found deletion of *Mll3* or *Mll4* also leads to a slow growth of mouse B16 melanoma cells during *in vitro* culture, which is notable starting from day 4 post cell seeding. There is no significant cell growth difference between wild-type and *Mll3* KO or *Mll4* KO B16 cells from day 1 to 3 after plating (Fig. R2A). LDH levels in the culture medium of *Mll3* KO or *Mll4* KO B16 cells in the absence of OT1 co-culture are close to the background level of growth medium and comparable across wild type, *Mll3* KO and *Mll4* KO B16 cells, indicating the viability of B16 cells is not affected by *Mll3* or *Mll4* deletion at early time points post cell seeding (Fig.R2B).

Figure R2. A. Growth curve of control, Mll3 KO or Mll4 KO B16 cells over a 5-day period of *in vitro* culture, determined by hemocytometer counting. B. LDH release assays of control, Mll3 KO or Mll4 KO B16 cells, analyzed every day over a 3-day period of *in vitro* culture.

5) Flow cytometry data related to Gzmb and IFN γ staining should be improved, as they appear not in a typical pattern.

Response: We appreciate the reviewer for this critique. When comparing the flow scatter plots of our IFN γ (Cat#61-7311-82, eBioscience) and GZMB (Cat#48-8898-82, eBioscience) signal with the representative flow cytometric plots provided in the respective antibody datasheet, our IFN γ and GZMB fluorescent signal looks weaker than the representative results shown by the manufacturers and does not separate into individual populations. This difference may result from the differential degree of T cell activation that is induced by distinct approaches and treatment conditions. For our *in vitro* OT-I T cell cytotoxicity assay, we did not pre-activate them with anti-CD3/CD28 antibodies before co-culture with B16 cells that ectopically express full-length OVA, which may lead to incomplete activation of these OT-I cells and thus lower intracellular levels of IFN γ and GZMB. We searched the literature in top-tier journals for flow cytometric detection of IFN γ and GZMB and found both of them are occasionally stained as a continuous distribution and do not split into separate clusters as well (Pan D, et al, 2018, *Science*, Fig. S10D; Wei J, et al, 2019, *Nature*, Fig. S3G, and S5C), which suggests that the activation and functional state of T cells may affect the distribution pattern of IFN γ and GZMB in flow cytometry analyses to some extent. Intriguingly, when we presented flow cytometry scatter plots of IFN γ and GZMB separately, better staining panels can be achieved and demonstrated a significant increase of IFN γ and GZMB levels in OT-I T cells incubated with *Mll3*- or *Mll4*-deleted B16 cells as compared to co-culture with control cells. To better present the flow scatter plots, we chose to present the IFN γ and GZMB staining signal in separate scatter plots instead of showing them in combination in the original figures. The separated scatter plots for IFN γ and GZMB are shown in Fig. 1h and supplementary fig. 1n and 1q.

6) A high E: T ratio of 10:1 was used in the killing assay, likely because OT1 cells are not activated and thus not very effective before being applied to tumor cell layers. In tumor masses, tumor cells usually numerically overwhelm T cells.

Response: We agree with the reviewer that the low cytotoxicity of OT-1 T cells and thus the high E: T ratio we used for *in vitro* tumor cell killing assay likely result from the incomplete activation of OT-1 T cells stimulated by B16 tumor cells ectopically expressing full-length Ova protein. As the activation status of OT-1 T cells is markedly affected by the cell surface level of MHC-1-bound SIINFEKL peptide, we postulated that the full-length Ova protein may be inefficiently processed and

presented by MHC-1 in B16 cells, which were not stimulated with IFN γ before co-culture with OT-1 T cells. In addition, our OT-1 T cells are not pre-activated with anti-CD3/CD28 beads, which are occasionally used in literature to boost T cell activation for *in vitro* cytotoxicity assay^{6,7}. One of the advantages of using less fully activated OT-1 cells in *in vitro* cytotoxicity assay is that the negative effects of immune regulators in tumor cells can be readily detected as highly activated T cells may mask the increased cytotoxicity of effector T cells elicited by loss of tumor cell immune suppressors.

7) When talking about “parental”, did the author refer to cells without any genetic modifications or cells also modified by CRISPR/Cas9 with control sgRNA? The use of “Parental” should be avoided if the latter is the case. Capital letters should be used when talking about proteins, for example MLL3, DNMT1 and so on.

Response: We appreciate the reviewer for pointing out the inappropriate description of control cells and the informal use of protein terms. Actually, the cells we used as controls have been transduced with lentivirus carrying an empty vector for sgRNA cloning and thus are not equal to the authentic parental cells that do not have any genetic manipulations. We replaced “parental” with “sgVector control” and symbols for all proteins have been capitalized throughout the revised manuscript.

8) In heatmaps of Figure 3, n=5 was claimed in the legends, but only two replicates can be clearly spotted from the figures.

Response: We apologize for the confusing description of RNA-seq replicates. “n=5” indicates that single cells from five sgVector, *Mll3* or *Mll4*-deficient B16 tumors were pooled for two independent flow cytometric sorting of GFP⁺ tumor cells. Total RNA was extracted from the two batches of sorted tumor cells and subject to total RNA-seq library preparation. To make it clear, we added more details about the preparation of the RNA-seq replicates in the section “RNA-seq and ChIP-seq Library preparation, Materials and Methods” (From lines 840-843).

9) Will the concurrent depletion of dsRNA and pyroptosis pathways completely rescue the growth defect of Mll3 and Mll4 KO tumors in immunocompetent mice?

Response: The reviewer raised a good question that needs to be addressed in our future study. However, from our RNA-seq data and the other known biochemical function of MLL3 and MLL4 in cellular regulation, such as DNA damage response^{8,9}, we can predict that simultaneous depletion of dsRNA and pyroptosis pathways is insufficient for fully rescuing the growth defects of *Mll3* KO and *Mll4* KO tumors in immunocompetent mice as the expression of many dsDNA sensors and components of MHC-I antigen presentation pathway are elevated in *Mll3*- and *Mll4*-deficient tumor cells (Fig.3e, 3h, supplementary Fig. 3g, and 3h). In support of our prediction, a recent study revealed that *Mll4* deletion increases the mutation load of liver tumors

and sensitizes them to immune checkpoint blockade treatment in immunocompetent mice¹⁰.

10) The cartoon in the last Figure should be modified, because naïve T cells are mainly primed in the lymphoid tissues and recruited to the tumor sites, rather than be primed locally in tumor masses, although sometimes local priming occurs for example in TLS.

Response: We appreciate the reviewer for pointing this out and have removed naïve T cells in the cartoon in the updated Fig. 7n.

Reviewer #4, expertise in pyroptosis and immunotherapy (Remarks to the Author):

In this manuscript, Ning, Huang and Lei et al. investigated how depletion of Mll3 and Mll4, two enhancer-associated H3K4 mono-methyltransferases, affects the tumor microenvironment and anti-tumor immunity. They present large amounts of data to claim that both dsRNA-interferon pathway and pyroptotic cell death are important for increased anti-tumor immune responses observed in Mll4 deficient tumors. However, some critical data regarding pyroptosis are not convincing, and some are missing to support their claims. Here are my concerns about this manuscript:

Response: We appreciate the reviewer for raising these constructive critiques to help us further improve the quality of this work.

Major concerns:

1) The author co-incubated CD8 OT-1 cells and B16-Ova in vitro to detect pyroptosis in B16-Ova cells with various genetic manipulations (Fig.4 and 5). These images should be the most convincing data because “seeing is believing.” However, the data presented is disappointed. Pyroptotic cells usually develop giant membrane balloons, but I basically can’t see any cells exhibiting pyroptotic morphology in Gsdmd-WT cells (Fig. 4, in four indicated cells, only one cell shows pyroptotic morphology but random necrosis can’t be ruled out in this cell) and in Mll4KO shGFP cells (Fig.5). The Mll4KO shGFP cells actually exhibit the morphology of apoptosis (shrunken cells with small blebbing).

Response: We thank the reviewer for bringing up this critical point. When re-examining morphological features of the indicated dying cells and comparing them to the typical pyroptotic morphology carefully, most cells indicated with arrowheads indeed do not manifest the typical pyroptotic morphology of cell swelling. This is consistent with cleavage and activation of only a small fraction of exogenous wild-type full-length GSDMD in B16 cells upon challenge with OT-I T cells, which implies a small percentage of target cells undergoing pyroptotic cell death (Fig.4f). It has now become increasingly clear that cytotoxic T lymphocytes can kill tumor cells through multiple cell death programs, including apoptosis, necroptosis, and the

recently recognized pyroptotic pathway¹¹⁻¹⁴. We think that T cell-triggered tumor cell death will not be constrained to pyroptosis only and activation of other forms of cell death program, such as apoptosis and necroptosis, will also occur during cytotoxic T cell attacks, as long as constituents of these cell death pathways and their upstream activating signals co-exist in the target tumor cells. For example, T and CAR-T cells can release granzymes to cleave executors of both apoptotic and pyroptotic pathways to induce distinct morphologic features of cell death¹³⁻¹⁶. In addition, apoptotic and pyroptotic morphologies can occur concomitantly in tumor cells when treated with chemotherapeutic drugs¹⁷⁻¹⁹. Moreover, we found it is a little bit subjective and hard to use morphologic features to qualitatively and quantitatively determine the paths of cell death in some cell types, especially when diverse cell death programs are concurrently activated and a mixture of various cell death features appear in those cells. Thus, we think it is more appropriate and objective to define the type of cell death through monitoring the activation status of terminal executors of cell death pathways, such as Gasdermin cleavage and MLKL phosphorylation as a maker for pyroptosis and necroptosis, respectively, as well as assessing the outcome of cell death, such as the release of cytosolic contents, LDH, and HMGB1, etc, into the growth medium. Furthermore, we also repeated the experiment in Fig. 4g and 5b with the extension of co-culture of target and effector cells to 12 hours to see if typical pyroptotic morphology can be induced in some target cells. As expected, we found ectopic expression of wild-type GSDMD or depletion of MLL4 can indeed trigger the formation of large ballooning bubbles in a few B16-Ova cells when co-cultured with antigen-specific T cells (Fig. R3).

Figure R3. A. B16-Ova cells expressing vehicle, N-terminally HA-tagged WT or D276A mutant *Gsdmd* were pre-loaded with fluorescent dye calcein AM and then co-cultured with OT-I CD8+ T cells at an E/T ratio of 10:1 for 12 hours. Representative images are shown with arrowheads indicating tumor cells undergoing pyroptosis. Scar bar: 20 μ M. B. Control or *Mll4*^{-/-} B16-Ova cells depleted for GSDMD were pre-loaded with fluorescent dye calcein AM and then co-cultured with OT-I CD8+ T cells at an E/T ratio of 10:1 for 12 hours. Representative images are shown with arrowheads indicating tumor cells undergoing pyroptosis. Scar bar: 20 μ M.

What makes it worse is that there is no cell death in any other B16 cells with low GSDMD expression in Fig.4 or Fig.5. But we know it is not the case. CD8 T OT-I cells will kill B16-Ova cells regardless of GSDM expression. The GSDMD may switch the types of cell death (e.g., from apoptosis to pyroptosis), but cell death should be observed in B16-Ova cells after the CD8 cell challenge.

Response: We agree with the reviewer that cytotoxic T cells will kill B16-Ova cells regardless of GSDM expression as components of other cell death pathways are expressed in this cell type. This is exactly what we saw in our cytotoxicity assay that reveals about 15% of cell death upon co-culture of vehicle vector-transduced B16-Ova cells with CD8 OT-I T cells as reflected in both calcein and LDH release experiments (Fig. 4g, 4h, 5b, and 5c). Without CD8 OT-I T cell challenge, the LDH level in supernatant of wild type, *Mil3*KO, and *Mil4* KO B16-Ova cells is very low during routine cell passages and close to the background level in complete cell growth medium. We also want to mention that CD8 OT-I T cells used in our in vitro cytotoxicity assay have not been pre-activated with anti-CD3/CD28 antibodies and thus are less cytotoxic compared to CD8 T cells that have been exposed to anti-CD3/CD28 antibodies before co-culture with target tumor cells. In addition, we expressed full-length Ova protein in B16 cells rather than pulse them with Ova peptide, which may lead to incomplete T-cell activation due to the inefficient processing and presentation of Ova antigen to cell-surface MHC I machinery and thus weak T-cell cytotoxicity. The use of less fully activated CD8 OT-I T cells in vitro cytotoxicity assay can help identification of tumor-cell intrinsic factors with mild immuno-stimulatory activities, otherwise, they might be missed out if using fully activated CD8 T cells with strong cytotoxicity.

Figure R4. LDH release assay of control, *Mil3* KO, and *Mil4* KO B14 cells during a 3-day period of *in vitro* culture. LDH level was expressed as mean \pm S.D from three technical replicates in one representative experiment.

2) In the immunoblots about GSDMD (Fig. S4C, S4D, S6J), there are GSDMD-NT fragments detected in all samples. It is surprising because GSDMD-NT generation by protease cleavage is an indicator of pyroptosis, and the data suggest that pyroptosis

occurs in all the cells. There must be something wrong with the data or the interpretation of the experiments (the GSDMD-NT band could be an unspecific band).

Response: We thank the reviewer for pointing out this question which might confuse the readers. In supplementary Fig. 4c, 4d, and 6k, immunoblots of GSDMD were performed on whole cell extracts of intact tumors that derive from mice inoculated with wild type, *Mil3*KO or *Mil4* KO B16 cells (supplementary Fig. 4c and 4d) or implanted with wild-type B16 cells and then treated with saline or DNMT inhibitor Decitabine (supplementary Fig. 6k). Tumor cells have not been sorted out and thus the levels of full-length and cleaved GSDMD shown in each blot actually reflect a sum of GSDMD expression and its activation from both tumor cells and surrounding immune cells in the tumor microenvironment. Low level of full-length and truncated GSDMD detected in sgVector B16 tumors in supplementary Fig.4c, 4d and in B16 tumors treated with saline in supplementary Fig. 6k may come from tumor-infiltrated macrophages and neutrophils, as it has been reported that CAR T cell granzyme B-mediated tumor cell death and bacterial infection can induce inflammatory caspase activation and GSDMD cleavage in macrophages and neutrophils, respectively^{15,20}. The elevated levels of full-length and cleaved GSDMD in *Mil3*KO or *Mil4* KO B16 tumors or B16 tumors treated with DNMTs inhibitor could be attributed to the transcriptional derepression of GSDMD expression in tumor cells upon loss of promoter DNA methylation that is triggered by *Mil3* or *Mil4* deletion or by direct DNMTs inhibition, which is what we want to propose in this study.

3) The mechanism underlying this CD8-triggered GSDMD pyroptosis is unclear and the claimed pyroptosis is not consistent with previous studies. The authors claim that GSDMD is activated by CD8 T cells, and the cleavage site is D276 (cleavage site of Caspase-1/4/5/11). CD8 T cells induce tumor cell death mainly by delivering granzymes. Previous studies indicate that GSDMB and GSDME are cleaved and activated by GzmA and GzmB, respectively. But the Gzm pathways may not fit the current case because GzmB is the only known granzyme that cuts after aspartic acid, but it has been proved that GSDMD is not a substrate of GzmB or any other Gzms (Ref.40 Fig S2C). Moreover, NK cells don't induce pyroptosis in cells expressing GSDMD (Ref.40), which is inconsistent with current data.

Response: The reviewer raised a excellent point that we have been thinking of since we noted GSDMD cleavage in *Mil4* KO B16 cells upon incubation with CD8 T cells. We initially thought GSDMD may act, like GSDMB and GSDME, as a direct substrate of a certain granzyme. This assumption was partially negated by a paper from Dr. Feng Shao's lab, demonstrating that recombinant GSDMD can't be proteolyzed by GZMs A, B, H, K, and M in *in vitro* protein cleavage assay¹³. That study also showed that ectopic expression of GSDMD does not increase pyroptosis of 293T cells when challenged with a human natural killer cell line NK-92MI. Though human and mice share several granzyme family members, both organisms have their specific granzymes, with *GzmH* present exclusively in human while other granzymes,

including *Gzms*, *C*, *D*, *E*, *F*, *G*, and *L*, are only found in mice²¹. Little is known about the biochemical function and enzymatic properties of these mice-specific granzymes at present. Our analysis of a publically available scRNA-seq dataset revealed detectable levels of these mice-specific granzymes in tumor-infiltrating CD8⁺ T cells, albeit at a much lower level than *Gzm A* and *Gzm B*²². Thus, one possibility is that the GSDMD cleavage may be mediated by these mice-specific granzymes when they are delivered into B16 cells upon incubation with mouse CD8 OT-1 T cells. Secondly, while perforin/granzyme pathway largely accounts for the CD8⁺ T cell cytotoxicity, TNF α /FASL signaling-induced caspase-8 activation and subsequent apoptotic cell death is also indispensable for efficient and complete clearance of virus-infected and transformed cells^{23,24}. Intriguingly, recent studies revealed that, besides the canonical and non-canonical inflammasome pathways, GSDMD activation can also be elicited by other mechanisms, such as caspase 8-mediated cleavage under conditions of TNF α -induced extrinsic apoptosis or inhibition of the kinase TAK1 or the inhibitor of apoptosis (IAP) by pharmacological agents or pathogen-derived effector proteins^{19,25-27}. Caspase-8 cuts GSDMD at the same position cleaved by inflammatory caspases, namely D276 and D277 in mice and human, respectively²⁷, which inspires us to propose that cleavage of exogenous GSDMD in B16 cells or endogenous GSDMD depressed in *Mil4* KO B16 cells may result from TNF α /FASL signaling-induced caspase-8 activation upon co-culture with CD8 OT-1 T cells. Lastly, by comparing the expression levels of known inflammasome components in wild-type and *Mil4* KO B16 cells, we found that *Mil4* loss leads to a marked increase in mRNA level of *IFI16*, which can act as a nuclear immune sensor for viral genome to trigger inflammasome assembly and caspase-1 activation²⁸. Our RNA-seq and immunofluorescent results revealed a striking increase of ERV transcripts (Fig, 3k, supplementary 3k, 3l, and 3m), some of which could be reverse-transcribed into DNAs for genomic integration²⁹. In light of these previous literature, we hypothesize that transcriptional upregulation of *IIF16* (known as *Ifi204* in mice) and its subsequent DNA sensing and inflammasome activation could also be involved in GSDMD cleavage in *Mil4* KO B16 cells when targeted by cytotoxic CD8 lymphocytes. We are testing these possibilities and related findings will be submitted as a separate manuscript.

Figure R5. A. Violin plot showing scRNA-seq expression of members of granzyme family in tumor-infiltrating CD8⁺ T cells. B. Histogram showing RNA-seq FPKM

values of known inflammasome sensors in control and *Mll4* KO B16 cells.

To support their claims, the authors need to:

a) Demonstrate that inflammasome-activated pyroptosis could occur in B16. Inflammasome-activated pyroptosis mainly occurs in macrophages and some epithelial cells, not in other cells, because inflammasome components are selectively expressed. I can't find any paper showing that B16 cells undergo inflammasome-activated pyroptosis. Therefore, the authors need to prove it. They may try some classical pyroptosis treatments (e.g., LPS+ATP and Poly dA:dT for canonical inflammasome; LPS electroporation for non-canonical inflammasome) to show that inflammasome assembly and pyroptosis occur in B16.

Response: We appreciate the reviewer for this constructive comment. As the reviewer stated, pyroptosis was indeed mainly studied in immune cells and some cell lines of epithelial origin in the pyroptosis field because of the tissue/cell type-restricted expression of inflammasome components and downstream pyroptotic executors. Our RNA-seq results are in line with previous literature and confirm very low or undetectable levels of known components in the pyroptotic pathway in wild-type B16 cells (Fig. 4a and R5B). However, we found that *MLL4* loss leads to a widespread reprogramming of enhancer epigenetic landscape and transcriptional derepression of a few pyroptotic genes, including *Gsdmd*, *Casp1*, and *Casp11* as well as the nuclear pathogen sensor *IFI204*²⁸ (Fig. 4a and R5B). As nearly all canonical inflammasome components remain undetectable even in the absence of *MLL4* expression, we speculate that stimulation with agents for canonical inflammasome activation will not cause pyroptotic cell death in both wild-type and *Mll4* KO B16 cells, which obviates conduction of the canonical inflammasome activation experiment as suggested by the reviewer.

Cytosolic LPS binds directly to caspase-4/5/11 to induce non-canonical inflammasome assembly and pyroptotic cell death in immune cells and other cell types that express caspase-4/5/11, such as HeLa, HT29, etc. Re-introduction of caspase-4/5/11 into 293T cells, which lack expression of non-canonical inflammasome components, causes massive pyroptotic cell death in response to cytosolic LPS exposure³⁰. Theoretically speaking, LPS electroporation will induce activation of caspase-4/5/11 and emergence of pyroptosis in any cell type with the expression of GSDMD and the non-canonical inflammatory caspases, irrespective of their tissue origin. To prove this hypothesis, we electroporated LPS into wild-type and *Mll4* KO B16 cells and found a typical pyroptotic morphology, GSDMD cleavage as well as extensive cell death are induced in *Mll4* KO but not wild-type B16 cells (Fig. R6), which agrees with the extremely low levels of inflammatory caspases and GSDMD in wild-type B16 cells but markedly increased upon *Mll4* deletion.

Figure R6. A. Morphology of control and *Mil4* KO B16 cells electroporated with or without LPS. Cells indicated with arrowheads are undergoing pyroptotic cell death. B. LDH release assays of control and *Mil4* KO B16 cells electroporated with or without LPS. C. Immunoblotting of GSDMD in control and *Mil4* KO B16 cells electroporated with or without LPS.

b) Show caspase-1 or caspase-11 activation (cleavage), PI uptake, pyroptotic morphology and other indicators of pyroptosis in B16 cells treated with CD8 T cells.

Response: We appreciate the reviewer for this suggestion. As our RNA-seq analyses show very low expression of inflammatory caspases and pyroptotic executors in wild-type B16 cells ((Fig. 4a and R5B), we thus agree with the reviewer and think that pyroptotic cell death will unlikely be induced in wild-type B16 cells upon co-culture with CD8 T cells. However, *Mil4* deletion decreases the expression of DNA methyltransferases and leads to a marked transcriptional derepression of *Casp1*, *Casp11*, and *Gsdmd* in B16 cells, which will sensitize *Mil4* KO B16 cells to pyroptosis induction when targeted by CD8⁺ T cells (Fig.5 and R3B). Our bioinformatic analyses of publically available datasets and experiments with a mouse model of melanoma indicate a vital role of GSDMD expression and cleavage at D276 in tumor immunity (Fig 5, S5, 7, and S7), suggesting the implication of GSDMD-mediated pyroptosis in augmenting anti-tumor immune response, though the detailed molecular mechanism for CD8⁺ T cell-triggered GSDMD cleavage in *Mil4* KO B16 cells is currently unknown.

c) Find out how CD8 T cells induce B16 pyroptosis.

Response: As the expression of both inflammatory caspases and gasdermin family members is lowly transcribed in wild-type B16 cells, we think pyroptotic cell death will unlikely be induced in wild-type B16 cells incubated with CD8 T cells. *Mil4* loss transcriptionally derepresses *Casp1*, *Casp11*, and *Gsdmd* in B16 cells and renders them to pyroptosis induction upon co-culture with CD8 T cells. We are currently unclear about the molecular mechanisms by which GSDMD is cleaved at D276 and

pyroptotic cell death is induced in *Mll4* KO B16 cells by CD8 T cell incubation. Based on the available literature and our transcriptomic profiling of wild-type and *Mll4* KO B16 cells, we plan to test three potential mechanisms that may account for CD8 T cell-elicited GSDMD cleavage and pyroptosis induction in *Mll4* KO B16 cells (see the reviewer's comment 3 for more detail). 1). whether mouse-specific granzymes cleave GSDMD and mediate pyroptosis induction in *Mll4* KO B16 cells. 2). the potential involvement of extrinsic apoptotic pathway and caspase-8 activation in GSDMD cleavage. 3). whether reverse transcription of ERVs and IFI204-mediated DNA sensing and CASP1 activation are involved in GSDMD cleavage and pyroptosis induction. The mechanisms responsible for GSDMD cleavage and pyroptotic induction of target tumor cells should become a separate manuscript.

Minor concerns:

1) As far as I know, the counterpart of human caspase-4 is mouse caspase-11, and there is no mouse caspase-4, or researchers usually don't use this name.

Response: Thanks to the reviewer for this comment and suggestion. We replaced *Casp4* with *Casp11* throughout all figures and the revised text.

2) The authors need to distinguish between human and mouse proteins. The statement in line 110-111 "The Gasdermin (Gsdm) family of pore-forming proteins that comprise Gsdma, Gsdmb, Gsdmc, Gsdmd, and Gsdme" is wrong. Human and mouse proteins share the same names, such as GSDMD; you can tell them using hGSDMD and mGSDMD. If referred to genes, use human GSDMD and mouse Gsdmd (italicized). More importantly, the human GSDM family contains GSDMA, GSDMB, GSDMC, GSDMD, GSDME and DFNB59. But mouse GSDM family includes Gsdma 1-3, Gsdmc 1-4, Gsdmd, Gsdme and Dfnb59. The authors studied the mouse GSDM family throughout the manuscript, but they never mentioned this information or never stated it correctly. In figure 4A and S4A, the authors must change the name Gsdma to Gsdma1, and Gsdmc to Gsdmc1.

Response: We are grateful to the reviewer for instructing us to use a more proper nomenclature of mouse and human gasdermin family proteins. All gene and protein symbols have been corrected and now follow the nomenclature throughout the figures and in the revised text.

3) Figure legend 1B, ET ratio 1:10 should be 10:1

Response: We thank the reviewer for pointing this out and have corrected it in the revised manuscript.

Reference

- 1 Alam, H. *et al.* KMT2D Deficiency Impairs Super-Enhancers to Confer a Glycolytic Vulnerability in Lung Cancer. *Cancer cell* **37**, 599-617 e597, doi:10.1016/j.ccell.2020.03.005 (2020).

- 2 Dhar, S. S. *et al.* MLL4 Is Required to Maintain Broad H3K4me3 Peaks and Super-Enhancers
at Tumor Suppressor Genes. *Molecular cell* **70**, 825-841 e826,
doi:10.1016/j.molcel.2018.04.028 (2018).
- 3 Ortega-Molina, A. *et al.* The histone lysine methyltransferase KMT2D sustains a gene
expression program that represses B cell lymphoma development. *Nature medicine* **21**,
1199-1208, doi:10.1038/nm.3943 (2015).
- 4 Zhang, J. *et al.* Disruption of KMT2D perturbs germinal center B cell development and
promotes lymphomagenesis. *Nature medicine* **21**, 1190-1198, doi:10.1038/nm.3940 (2015).
- 5 Guo, C. *et al.* KMT2D maintains neoplastic cell proliferation and global histone H3 lysine 4
monomethylation. *Oncotarget* **4**, 2144-2153, doi:10.18632/oncotarget.1555 (2013).
- 6 Lawson, K. A. *et al.* Functional genomic landscape of cancer-intrinsic evasion of killing by T
cells. *Nature* **586**, 120-126, doi:10.1038/s41586-020-2746-2 (2020).
- 7 Pan, D. *et al.* A major chromatin regulator determines resistance of tumor cells to T
cell-mediated killing. *Science* **359**, 770-775, doi:10.1126/science.aao1710 (2018).
- 8 Santos, M. A. *et al.* DNA-damage-induced differentiation of leukaemic cells as an anti-cancer
barrier. *Nature* **514**, 107-111, doi:10.1038/nature13483 (2014).
- 9 Lv, S. *et al.* Loss of KMT2D induces prostate cancer ROS-mediated DNA damage by
suppressing the enhancer activity and DNA binding of antioxidant transcription factor FOXO3.
Epigenetics **14**, 1194-1208, doi:10.1080/15592294.2019.1634985 (2019).
- 10 Wang, G. *et al.* CRISPR-GEMM pooled mutagenic screening identifies KMT2D as a major
modulator of immune checkpoint blockade. *Cancer discovery*,
doi:10.1158/2159-8290.CD-19-1448 (2020).
- 11 Newton, K. & Manning, G. Necroptosis and Inflammation. *Annual review of biochemistry* **85**,
743-763, doi:10.1146/annurev-biochem-060815-014830 (2016).
- 12 Tummers, B. & Green, D. R. Caspase-8: regulating life and death. *Immunological reviews* **277**,
76-89, doi:10.1111/imr.12541 (2017).
- 13 Zhou, Z. *et al.* Granzyme A from cytotoxic lymphocytes cleaves GSDMB to trigger pyroptosis
in target cells. *Science* **368**, doi:10.1126/science.aaz7548 (2020).
- 14 Zhang, Z. *et al.* Gasdermin E suppresses tumour growth by activating anti-tumour immunity.
Nature **579**, 415-420, doi:10.1038/s41586-020-2071-9 (2020).
- 15 Liu, Y. *et al.* Gasdermin E-mediated target cell pyroptosis by CAR T cells triggers cytokine
release syndrome. *Science immunology* **5**, doi:10.1126/sciimmunol.aax7969 (2020).
- 16 Voskoboinik, I., Whisstock, J. C. & Trapani, J. A. Perforin and granzymes: function,
dysfunction and human pathology. *Nature reviews. Immunology* **15**, 388-400,
doi:10.1038/nri3839 (2015).
- 17 Wang, Y. *et al.* Chemotherapy drugs induce pyroptosis through caspase-3 cleavage of a
gasdermin. *Nature* **547**, 99-103, doi:10.1038/nature22393 (2017).
- 18 Rogers, C. *et al.* Cleavage of DFNA5 by caspase-3 during apoptosis mediates progression to
secondary necrotic/pyroptotic cell death. *Nature communications* **8**, 14128,
doi:10.1038/ncomms14128 (2017).
- 19 Chen, K. W. *et al.* Extrinsic and intrinsic apoptosis activate pannexin-1 to drive NLRP3
inflammasome assembly. *The EMBO journal* **38**, doi:10.15252/emboj.2019101638 (2019).
- 20 Chen, K. W. *et al.* Noncanonical inflammasome signaling elicits gasdermin D-dependent
neutrophil extracellular traps. *Science immunology* **3**, doi:10.1126/sciimmunol.aar6676

- (2018).
- 21 Grossman, W. J. *et al.* The orphan granzymes of humans and mice. *Current opinion in immunology* **15**, 544-552, doi:10.1016/s0952-7915(03)00099-2 (2003).
- 22 Gubin, M. M. *et al.* High-Dimensional Analysis Delineates Myeloid and Lymphoid Compartment Remodeling during Successful Immune-Checkpoint Cancer Therapy. *Cell* **175**, 1014-1030 e1019, doi:10.1016/j.cell.2018.09.030 (2018).
- 23 Golstein, P. & Griffiths, G. M. An early history of T cell-mediated cytotoxicity. *Nature reviews. Immunology* **18**, 527-535, doi:10.1038/s41577-018-0009-3 (2018).
- 24 Ratner, A. & Clark, W. R. Role of TNF-alpha in CD8+ cytotoxic T lymphocyte-mediated lysis. *Journal of immunology* **150**, 4303-4314 (1993).
- 25 Orning, P. *et al.* Pathogen blockade of TAK1 triggers caspase-8-dependent cleavage of gasdermin D and cell death. *Science* **362**, 1064-1069, doi:10.1126/science.aau2818 (2018).
- 26 Demarco, B. *et al.* Caspase-8-dependent gasdermin D cleavage promotes antimicrobial defense but confers susceptibility to TNF-induced lethality. *Science advances* **6**, doi:10.1126/sciadv.abc3465 (2020).
- 27 Sarhan, J. *et al.* Caspase-8 induces cleavage of gasdermin D to elicit pyroptosis during Yersinia infection. *Proceedings of the National Academy of Sciences of the United States of America* **115**, E10888-E10897, doi:10.1073/pnas.1809548115 (2018).
- 28 Kerur, N. *et al.* IFI16 acts as a nuclear pathogen sensor to induce the inflammasome in response to Kaposi Sarcoma-associated herpesvirus infection. *Cell host & microbe* **9**, 363-375, doi:10.1016/j.chom.2011.04.008 (2011).
- 29 Zhang, S. M. *et al.* KDM5B promotes immune evasion by recruiting SETDB1 to silence retroelements. *Nature* **598**, 682-687, doi:10.1038/s41586-021-03994-2 (2021).
- 30 Shi, J. *et al.* Inflammatory caspases are innate immune receptors for intracellular LPS. *Nature* **514**, 187-192, doi:10.1038/nature13683 (2014).

Reviewers' Comments:

Reviewer #1:

Remarks to the Author:

The authors have satisfied all of my concerns and made the appropriate recommended edits and additions, so I have no further comments and believe the paper is ready for acceptance.

Reviewer #3:

Remarks to the Author:

The authors have addressed my questions. Overall, it is a good fit for Nature Communications.

Reviewer #4:

Remarks to the Author:

I agree with the authors that CD8 T cells may induce multiple cell death including apoptosis, pyroptosis, necroptosis and ferroptosis. Therefore, the authors need to be cautious about CD8 T cell-induced pyroptosis claims. Single marker-like morphology is not enough to determine if cell death is pyroptosis. For example, pyroptosis and necroptosis share similar morphology (giant membrane balloon) because they are both programmed necrosis. Neither LDH/HMGB1 release nor GSDMD cleavage is enough because LDH/HMGB1 release is not a unique feature of pyroptosis (shared by other programmed necrosis and necrosis), and GSDMD cleavage isn't always an indicator of cell death. In the scenario of cell hyperactivation, GSDMD is activated and cleaved but cells survive. In the case of GSDME, caspase-3 triggered GSDME cleavage doesn't always lead to pyroptosis, but if the GSDME level is low, apoptosis occurs first and GSDME is responsible for subsequent secondary necrosis. Therefore, as the authors state, we should objectively define the type of cell death according to the experimental data. In my opinion, the data provided by the authors can't support their pyroptosis claim, not only because of the lack of pyroptotic morphology but also because of the complexity of CD8 T cells-induced cell death. The authors do provide Figure R3 to support their claims, in which the dead cells show giant membrane balloons at 12 hrs. But one should know that any dead cells would develop secondary necrosis, exhibiting the same morphology as pyroptosis, after such a long incubation. It is more likely that GSDMD in B16, like a low level of GSDME, mediates secondary necrosis as suggested by apoptotic morphology at 6 hr (Fig 5b) and necrotic morphology at 12 hr (Fig R3).

To further clarify, the authors may need to check if apoptosis, pyroptosis, necroptosis and ferroptosis occur at the same time (e.g., MLKL phosphorylation...) and use inhibitors of those cell death modalities (e.g., disulfiram for pyroptosis, nec-1s for necroptosis...) to see if any inhibitor could disrupt the observed phenotypes.

REVIEWER COMMENTS

Reviewer #1 (Remarks to the Author):

The authors have satisfied all of my concerns and made the appropriate recommended edits and additions, so I have no further comments and believe the paper is ready for acceptance.

Reviewer #3 (Remarks to the Author):

The authors have addressed my questions. Overall, it is a good fit for Nature Communications.

Reviewer #4 (Remarks to the Author):

I agree with the authors that CD8 T cells may induce multiple cell death including apoptosis, pyroptosis, necroptosis and ferroptosis. Therefore, the authors need to be cautious about CD8 T cell-induced pyroptosis claims. Single marker-like morphology is not enough to determine if cell death is pyroptosis. For example, pyroptosis and necroptosis share similar morphology (giant membrane balloon) because they are both programmed necrosis. Neither LDH/HMGB1 release nor GSDMD cleavage is enough because LDH/HMGB1 release is not a unique feature of pyroptosis (shared by other programmed necrosis and necrosis), and GSDMD cleavage isn't always an indicator of cell death. In the scenario of cell hyperactivation, GSDMD is activated and cleaved but cells survive. In the case of GSDME, caspase-3 triggered GSDME cleavage doesn't always lead to pyroptosis, but if the GSDME level is low, apoptosis occurs first and GSDME is responsible for subsequent secondary necrosis. Therefore, as the authors state, we should objectively define the type of cell death according to the experimental data. In my opinion, the data provided by the authors can't support their pyroptosis claim, not only because of the lack of pyroptotic morphology but also because of the complexity of CD8 T cells-induced cell death. The authors do provide Figure R3 to support their claims, in which the dead cells show giant membrane balloons at 12 hrs. But one should know that any dead cells would develop secondary necrosis, exhibiting the same morphology as pyroptosis, after such a long incubation. It is more likely that GSDMD in B16, like a low level of GSDME, mediates secondary necrosis as suggested by apoptotic morphology at 6 hr (Fig 5b) and necrotic morphology at 12 hr (Fig R3).

To further clarify, the authors may need to check if apoptosis, pyroptosis, necroptosis and ferroptosis occur at the same time (e.g., MLKL phosphorylation...) and use inhibitors of those cell death modalities (e.g., disulfiram for pyroptosis, nec-1s for necroptosis...) to see if any inhibitor could disrupt the observed phenotypes.

Response: We appreciate the reviewer for this critique and the constructive suggestion. Though cytotoxic T lymphocytes can trigger tumor cell death in diverse forms, the specific type of cell death induced is cell-type dependent and relies on the expression status and the activity of the components that constitute each cell death pathway in tumor cells. The ballooning membrane morphology observed in *Mil4*KO B16 cells upon co-culture with antigen-specific CD8⁺ T cells is unlikely caused by necroptosis as the expression of *Ripk3*, a central component in the necroptotic pathway, is undetectable in both wild-type and *Mil4*-ablated B16 cells as demonstrated in our

RNA-seq analyses (Fig.4a).

To determine whether the T-cell triggered pyroptotic phenotype of *Mll4*KO B16 cells is related to apoptosis, necroptosis, pyroptosis, and ferroptosis, Ova-expressing control and *Mll4*-ablated B16 cells were incubated with OT-I T cells in the presence of pharmacological agents inhibiting each of the above cell death pathways, respectively. Targeting pan-caspase by zVAD-fmk or inhibiting the pore-forming activity of GSDMD by Disulfiram could largely prevent the appearance of the swelling membrane morphology and markedly attenuates the extracellular release of LDH in *Mll4*-ablated B16 cells while suppression of either ferroptosis by Ferrostatin-1 or necroptosis by Necrostatin-1, does not show any noticeable impact on both plasma membrane swelling and LDH release as observed in DMSO-treated control *Mll4*KO B16 cells. These results indicate that caspases and GSDMD are indispensable for the induction of the observed giant membrane swelling morphology in the *Mll4*-ablated B16 cells when targeted by cytotoxic T lymphocytes. Moreover, these findings also support one of our previous speculations for the involvement of caspases in GSDMD cleavage and the subsequent pyroptotic induction in *Mll4*-ablated B16 cells.

Figure R1. Morphology of Ova-expressing control and *Mll4*-ablated B16 cells co-cultured with OT-I CD8⁺T cells at an E/T ratio of 10:1 for 12 hours in the presence of indicated pharmacologic agents targeting GSDMD (Disulfiram, 60 μ M), pan-caspases (Z-VAD-FMK, 20 μ M), ferroptosis (Ferrostatin-1, 2 μ M) and necroptosis (Necrostatin-1, 10 μ M), respectively. Representative images are shown with arrowheads indicating tumor cells undergoing pyroptosis. Scar bar: 10 μ m. B. Cells were treated as in (A) and LDH levels in co-culture supernatants of cells with indicated treatment were measured and shown as mean \pm S.D from technical triplicates of a representative experiment.

Pyroptosis is one type of necrotic cell death that was re-defined by Dr. Feng Shao's group as Gasdermin-mediated programmed necrosis. Depending on their expression levels in cells, gasdermins can either trigger primary necrosis in healthy cells to act as a predominant way of cell death (definition of pyroptosis) or mediate secondary necrosis in apoptotic cells if the dead cells are not scavenged in time. Our RNA-seq and immunoblotting experiments were performed on the

bulk cells, the increased levels of *Gsdmd*, *Casp1*, and *casp11* represent a sum of their depression from bulk *Mll4*-ablated B16 cells. Due to the inherent heterogeneity of tumor cells, the degree of epigenetic derepression of these factors after *MLL4* depletion may be not uniform and can vary among different single cells, leading to variable levels of GSDMD, CASP1, and CASP11 in individual *Mll4*-ablated single cells. Thus, we agree with the reviewer and think that the cytotoxic T lymphocytes-induced giant membrane balloon observed in *Mll4* KO B16 cells may be a consequence of both pyroptosis in GSDMD^{high} cells and secondary necrosis in GSDMD^{low} cells. Though pyroptosis and gasdermin-mediated secondary necrosis occur at different stages of cell death, both of them are programmed necrosis and dependent on gasdermin cleavage and activation. Thus, based on the molecular definition of pyroptosis by Dr. Feng Shao, the gasdermin-mediated secondary necrosis could also be termed pyroptosis in some sense. The common outcome of pyroptosis and secondary necrosis is to release cytosolic damage-associated molecular patterns (DAMPs) to trigger an inflammatory response, which is what we want to propose as one of the critical molecular bases underlying the augmented anti-tumor response observed in *Mll4*-ablated tumors. We added a paragraph to bring out this point at the end of the Discussion section in the new version of the manuscript (Lines 708 to 720).

Reviewers' Comments:

Reviewer #4:

Remarks to the Author:

I don't have further questions about the manuscript.